# STARC: A General Framework For Quantifying Differences Between Reward Functions

**Joar Skalse**
Department of Computer Science
Future of Humanity Institute
Oxford University
joar.skalse@cs.ox.ac.uk

**Lucy Farnik**
University of Bristol
Bristol AI Safety Centre
lucy.farnik@bristol.ac.uk

**Sumeet Ramesh Motwani**
Berkeley Artificial Intelligence Research
University of California, Berkeley
motwani@berkeley.edu

**Erik Jenner**
Berkeley Artificial Intelligence Research
University of California, Berkeley
jenner@berkeley.edu

**Adam Gleave**
FAR AI, Inc.
adam@far.ai

**Alessandro Abate**
Department of Computer Science
Oxford University
aabate@cs.ox.ac.uk

## Abstract

In order to solve a task using reinforcement learning, it is necessary to first formalise the goal of that task as a *reward function*. However, for many real-world tasks, it is very difficult to manually specify a reward function that never incentivises undesirable behaviour. As a result, it is increasingly popular to use *reward learning algorithms*, which attempt to *learn* a reward function from data. However, the theoretical foundations of reward learning are not yet well-developed. In particular, it is typically not known when a given reward learning algorithm with high probability will learn a reward function that is safe to optimise. This means that reward learning algorithms generally must be evaluated empirically, which is expensive, and that their failure modes are difficult to anticipate in advance. One of the roadblocks to deriving better theoretical guarantees is the lack of good methods for *quantifying* the difference between reward functions. In this paper we provide a solution to this problem, in the form of a class of pseudometrics on the space of all reward functions that we call STARC (STAndardised Reward Comparison) metrics. We show that STARC metrics induce both an upper and a lower bound on worst-case regret, which implies that our metrics are tight, and that any metric with the same properties must be bilipschitz equivalent to ours. Moreover, we also identify a number of issues with reward metrics proposed by earlier works. Finally, we evaluate our metrics empirically, to demonstrate their practical efficacy. STARC metrics can be used to make both theoretical and empirical analysis of reward learning algorithms both easier and more principled.

## 1 Introduction

To solve a sequential decision-making task with reinforcement learning or automated planning, we must first formalise that task using a reward function (Sutton & Barto, 2018; Russell & Norvig, 2020). However, for many tasks, it is extremely difficult to manually specify a reward function that captures the task in the intended way. To resolve this issue, it is increasingly popular to use *reward learning*, which attempts to *learn* a reward function from data. There are many techniques for doing this. For example, it is possible to use preferences between trajectories (e.g. Christiano et al., 2017), expert demonstrations (e.g. Ng & Russell, 2000), or a combination of the two (e.g. Ibarz et al., 2018).

To evaluate a reward learning method, we must *quantify* the *difference* between the learnt reward function and the underlying true reward function. However, doing this is far from straightforward. A simple method might be to measure their $L_2$-distance. However, this is unsatisfactory, because two reward functions can have a large $L_2$-distance, even if they induce the *same* ordering of policies, or a small $L_2$-distance, even if they induce the *opposite* ordering of policies.[1] Another option is to evaluate the learnt reward function on a *test set*. However, this is also unsatisfactory, because it can only guarantee that the learnt reward function is accurate on a given data distribution, and when the reward function is *optimised* we necessarily incur a *distributional shift* (after which the learnt reward function may no longer match the true reward function). Yet another option is to optimise the learnt reward function, and evaluate the obtained policy according to the true reward function. However, this is also unsatisfactory, both because it is very expensive, and because it makes it difficult to separate issues with the policy optimisation process from issues with the reward learning algorithm. Moreover, because this method is purely empirical, it cannot be used for theoretical work. These issues make it challenging to evaluate reward learning algorithms in a way that is principled and robust. This in turn makes it difficult to anticipate in what situations a reward learning algorithm might fail, or what their failure modes might look like. It also makes it difficult to compare different reward learning algorithms against each other, without getting results that may be heavily dependent on the experimental setup. These issues limit the applicability of reward learning in practice.

In this paper, we introduce STAndardised Reward Comparison (STARC) metrics, which is a family of *pseudometrics* that quantify the difference between reward functions in a principled way. Moreover, we demonstrate that STARC metrics enjoy strong theoretical guarantees. In particular, we show that STARC metrics induce an upper bound on the worst-case regret that can be induced under arbitrary policy optimisation, which means that a small STARC distance guarantees that two reward functions behave in a similar way. Moreover, we also demonstrate that STARC metrics induce a *lower* bound on worst-case regret. This has the important consequence that any reward function distance metric which induces both an upper and a lower bound on worst-case regret must be bilipschitz equivalent to STARC metrics, which in turn means that they (in a certain sense) are unique. In particular, we should not expect to be able to improve on them in any substantial way. In addition to this, we also evaluate STARC metrics experimentally, and demonstrate that their theoretical guarantees translate into compelling empirical performance. STARC metrics are cheap to compute, which means that they can be used for empirical evaluation of reward learning algorithms. Moreover, they can be calculated from a closed-form expression, which means that they are also suitable for use in theoretical analysis. As such, STARC metrics enable us to evaluate reward learning methods in a way that is both easier and more theoretically principled than relevant alternatives. Our work thus contributes towards building a more rigorous foundation for the field of reward learning.

## 1.1 RELATED WORK

There are two existing papers that study the problem of how to quantify the difference between reward functions. The first is Gleave et al. (2020), which proposes a distance metric that they call Equivalent-Policy Invariant Comparison (EPIC). They show that the EPIC-distance between two reward functions induces a regret bound for optimal policies. The second paper is Wulfe et al. (2022), which proposes a distance metric that they call Dynamics-Aware Reward Distance (DARD). Unlike EPIC, DARD incorporates information about the transition dynamics of the environment. This means that DARD might give a tighter measurement, in situations where the transition dynamics are known. Unlike Gleave et al. (2020), they do not derive any regret bound for DARD.

Our work extends the work by Gleave et al. (2020) and Wulfe et al. (2022) in several important ways. First of all, Wulfe et al. (2022) do not provide any regret bounds, which is unsatisfactory for theoretical work, and the upper regret bound that is provided by Gleave et al. (2020) is both weaker and less general than ours. In particular, their bound only considers optimal policies, whereas our bound covers all pairs of policies (with optimal policies being a special case). Moreover, we also argue that Gleave et al. (2020) have chosen to quantify regret in a way that fails to capture what we care about in practice. In Appendix A, we provide an extensive theoretical analysis of EPIC, and show

---

[1]For example, given an arbitrary reward function $R$ and an arbitrary constant $c$, we have that $R$ and $c \cdot R$ have the same ordering of policies, even though their $L_2$-distance may be arbitrarily large. Similarly, for any $\epsilon$, we have that $\epsilon \cdot R$ and $-\epsilon \cdot R$ have the opposite ordering of policies, unless $R$ is constant, even though their $L_2$-distance may be arbitrarily small.

that it lacks many of the important theoretical guarantees enjoyed by STARC metrics. In particular, we demonstrate that EPIC fails to induce either an upper or lower bound on worst-case regret (as we define it). We also include an extensive discussion and criticism of DARD in Appendix B. Moreover, in Section 4, we provide experimental data that shows that STARC metrics in practice can have a much tighter correlation with worst-case regret than both EPIC and DARD. This means that STARC metrics both can attain better empirical performance *and* give stronger theoretical guarantees than the pseudometrics proposed by earlier work.

It is important to note that EPIC is designed to be independent of the environment dynamics, whereas both STARC and DARD depend on the transition dynamics. This issue is discussed in Section 2.3.

The question of what happens if one reward function is optimised instead of a different reward function is considered by many previous works. A notable example is Ng et al. (1999), which shows that if two reward functions differ by a type of transformation they call *potential shaping*, then they have the same optimal policies in all environments. Potential shaping is also studied by e.g. Jenner et al. (2022). Another example is Skalse et al. (2022b), which shows that if two reward functions $R_1$, $R_2$ have the property that there are no policies $\pi_1, \pi_2$ such that $J_1(\pi_1) > J_1(\pi_2)$ and $J_2(\pi_1) < J_2(\pi_2)$, then either $R_1$ and $R_2$ induce the same ordering of policies, or at least one of them assigns the same reward to all policies. Zhuang & Hadfield-Menell (2021) consider proxy rewards that depend on a strict subset of the features which are relevant to the true reward, and then show that optimising such a proxy in some cases may be arbitrarily bad, given certain assumptions. Skalse et al. (2022a) derive necessary and sufficient conditions for when two reward functions are equivalent, for the purposes of computing certain policies or other mathematical objects. Also relevant is Everitt et al. (2017), which studies the related problem of reward corruption, and Pan et al. (2022), which considers natural choices of proxy rewards for several environments. Unlike these works, we are interested in the question of *quantifying* the difference between reward functions.

## 1.2 Preliminaries

A *Markov Decision Processes* (MDP) is a tuple $(\mathcal{S}, \mathcal{A}, \tau, \mu_0, R, \gamma)$ where $\mathcal{S}$ is a set of *states*, $\mathcal{A}$ is a set of *actions*, $\tau : \mathcal{S} \times \mathcal{A} \to \Delta(\mathcal{S})$ is a *transition function*, $\mu_0 \in \Delta(\mathcal{S})$ is an *initial state distribution*, $R : \mathcal{S} \times \mathcal{A} \times \mathcal{S} \to \mathbb{R}$ is a *reward function*, and $\gamma \in (0, 1)$ is a *discount rate*. A *policy* is a function $\pi : \mathcal{S} \to \Delta(\mathcal{A})$. A *trajectory* $\xi = \langle s_0, a_0, s_1, a_1 \dots \rangle$ is a possible path in an MDP. The *return function G* gives the cumulative discounted reward of a trajectory, $G(\xi) = \sum_{t=0}^{\infty} \gamma^t R(s_t, a_t, s_{t+1})$, and the *evaluation function J* gives the expected trajectory return given a policy, $J(\pi) = \mathbb{E}_{\xi \sim \pi}[G(\xi)]$. A policy maximising $J$ is an *optimal policy*. The *value function* $V^\pi : \mathcal{S} \to \mathbb{R}$ of a policy encodes the expected future discounted reward from each state when following that policy. We use $\mathcal{R}$ to refer to the set of all reward functions. When talking about multiple rewards, we give each reward a subscript $R_i$, and use $J_i$, $G_i$, and $V_i^\pi$, to denote $R_i$'s evaluation function, return function, and $\pi$-value function.

In this paper, we assume that all states are reachable under $\tau$ and $\mu_0$. Note that if this is not the case, then all unreachable states can simply be removed from $\mathcal{S}$. Our *theoretical results* also assume that $\mathcal{S}$ and $\mathcal{A}$ are finite. However, STARC metrics can still be computed in continuous environments.

Given a set $X$, a function $d : X \times X \to \mathbb{R}$ is called a *pseudometric* if $d(x_1, x_1) = 0$, $d(x_1, x_2) \geqslant 0$, $d(x_1, x_2) = d(x_2, x_1)$, and $d(x_1, x_3) \leqslant d(x_1, x_2) + d(x_2, x_3)$, for all $x_1, x_2, x_2 \in X$. Given two pseudometrics $d_1, d_2$ on $X$, if there are constants $\ell, u$ such that $\ell \cdot d_1(x_1, x_2) \leqslant d_2(x_1, x_2) \leqslant u \cdot d_1(x_1, x_2)$ for all $x_1, x_2 \in X$, then $d_1$ and $d_2$ are *bilipschitz equivalent*. Given a vector space $V$, a function $n : V \to \mathbb{R}$ is a *norm* if $n(v_1) \geqslant 0$, $n(v_1) = 0 \iff v_1 = 0$, $n(c \cdot v_1) = |c| \cdot n(v_1)$, and $n(v_1 - v_2) \leqslant n(v_1) + n(v_2)$ for all $v_1, v_2 \in V$, $c \in \mathbb{R}$. Given a norm $n$, we can define a (pseudo)metric $m$ as $m(x, y) = n(|x - y|)$. In a mild abuse of notation, we will often denote this metric using $n$ directly, so that $n(x, y) = n(|x - y|)$. For any $p \in \mathbb{N}$, $L_p$ is the norm given by $L_p(v) = (\sum |v_i|^p)^{1/p}$. A norm $n$ is a *weighted* version of $n'$ if $n = n' \circ M$ for a diagonal matrix $M$.

We will use *potential shaping*, which was first introduced by Ng et al. (1999). First, a *potential function* is a function $\Phi : \mathcal{S} \to \mathbb{R}$. Given a discount $\gamma$, we say that $R_1$ and $R_2$ differ by *potential shaping* if for some potential $\Phi$, we have that $R_2(s, a, s') = R_1(s, a, s') + \gamma \cdot \Phi(s') - \Phi(s)$. We also use $S'$-*redistribution* (as defined by Skalse et al., 2022a). Given a transition function $\tau$, we say that $R_1$ and $R_2$ differ by $S'$-redistribution if $\mathbb{E}_{S' \sim \tau(s,a)}[R_2(s, a, S')] = \mathbb{E}_{S' \sim \tau(s,a)}[R_1(s, a, S')]$. Finally, we say that $R_1$ and $R_2$ differ by positive linear scaling if $R_2(s, a, s') = c \cdot R_1(s, a, s')$ for some positive constant $c$. We will also combine these transformations. For example, we say that $R_1$

and $R_2$ differ by potential shaping and $S'$-redistribution if it is possible to produce $R_2$ from $R_1$ by applying potential shaping and $S'$-redistribution (in any order). The cases where $R_1$ and $R_2$ differ by (for example) potential shaping and positive linear scaling, etc, are defined analogously. Finally, we will use the following result, proven by Skalse & Abate (2023) in their Theorem 2.6:

**Proposition 1.** $(S, A, \tau, \mu_0, R_1, \gamma)$ *and* $(S, A, \tau, \mu_0, R_2, \gamma)$ *have the same ordering of policies if and only if* $R_1$ *and* $R_2$ *differ by potential shaping, positive linear scaling, and* $S'$*-redistribution.*

The "ordering of policies" is the ordering induced by the policy evaluation function $J$.

EPIC (Gleave et al., 2020) is defined relative to a distribution $\mathcal{D}_S$ over $S$ and a distribution $\mathcal{D}_A$ over $A$, which must give support to all states and actions. It is computed in several steps. First, let $C^{\text{EPIC}} : \mathcal{R} \to \mathcal{R}$ be the function where $C^{\text{EPIC}}(R)(s, a, s')$ is equal to

$$R(s, a, s') + \mathbb{E}[\gamma R(s', A, S') - R(s, A, S') - \gamma R(S, A, S')],$$

where $S, S' \sim \mathcal{D}_S$ and $A \sim \mathcal{D}_A$. Note that $S$ and $S'$ are sampled independently. Next, let the "Pearson distance" between two random variables $X$ and $Y$ be defined as $\sqrt{(1 - \rho(X, Y))/2}$, where $\rho$ denotes the Pearson correlation. Then the EPIC-distance $D^{\text{EPIC}}(R_1, R_2)$ is defined to be the Pearson distance between $C^{\text{EPIC}}(R_1)(S, A, S')$ and $C^{\text{EPIC}}(R_2)(S, A, S')$, where again $S, S' \sim \mathcal{D}_S$ and $A \sim \mathcal{D}_A$.[2] Note that $D^{\text{EPIC}}$ is implicitly parameterised by $\mathcal{D}_S$ and $\mathcal{D}_A$.

To better understand how EPIC works, it is useful to know that it can be equivalently expressed as

$$D^{\text{EPIC}}(R_1, R_2) = \frac{1}{2} \cdot L_{2,\mathcal{D}} \left( \frac{C^{\text{EPIC}}(R_1)}{L_{2,\mathcal{D}}(C^{\text{EPIC}}(R_1))}, \frac{C^{\text{EPIC}}(R_2)}{L_{2,\mathcal{D}}(C^{\text{EPIC}}(R_2))} \right),$$

where $L_{2,\mathcal{D}}$ is a weighted $L_2$-norm. For details, see Appendix E. Here $C^{\text{EPIC}}$ maps all reward functions that differ by potential shaping to a single representative in their equivalence class. This, combined with the normalisation step, ensures that reward functions which only differ by potential shaping and positive linear scaling have distance 0 under $D^{\text{EPIC}}$.

DARD (Wulfe et al., 2022) is also defined relative to a distribution $\mathcal{D}_S$ over $S$ and a distribution $\mathcal{D}_A$ over $A$, which must give support to all actions and all reachable states, but it also requires a transition function $\tau$. Let $C^{\text{DARD}} : \mathcal{R} \to \mathcal{R}$ be the function where $C^{\text{DARD}}(R)(s, a, s')$ is

$$R(s, a, s') + \mathbb{E}[\gamma R(s', A, S'') - R(s, A, S') - \gamma R(S', A, S'')],$$

where $A \sim \mathcal{D}_A$, $S' \sim \tau(s, A)$, and $S'' \sim \tau(s', A)$. Then the DARD-distance $D^{\text{DARD}}(R_1, R_2)$ is defined to be the Pearson distance between $C^{\text{DARD}}(R_1)(S, A, S')$ and $C^{\text{DARD}}(R_2)(S, A, S')$, where again $S, S' \sim \mathcal{D}_S$ and $A \sim \mathcal{D}_A$. Note that $D^{\text{DARD}}$ is parameterised by $\mathcal{D}_S$, $\mathcal{D}_A$, and $\tau$.

## 2 STARC METRICS

In this section we formally define STARC metrics, and provide several examples of such metrics.

### 2.1 A FORMAL DEFINITION OF STARC METRICS

STARC metrics are defined relative to an environment, consisting of a set of states $S$, a set of actions $A$, a transition function $\tau$, an initial state distribution $\mu_0$, and a discount factor $\gamma$. This means that many of our definitions and theorems are implicitly parameterised by these objects, even when this dependency is not spelled out explicitly. Our results hold for any choice of $S$, $A$, $\tau$, $\mu_0$, and $\gamma$, as long as they satisfy the assumptions given in Section 1.2. See also Section 2.3.

STARC metrics are computed in several steps, where the first steps collapse certain equivalence classes in $\mathcal{R}$ to a single representative, and the last step measures a distance. The reason for this is that two distinct reward functions can share the exact same preferences between all policies. When this is the case, we want them to be treated as equivalent. This is achieved by standardising the reward functions in various ways before the distance is finally measured. First, recall that neither potential shaping nor $S'$-redistribution affects the policy ordering in any way. This motivates the first step:

---

[2]Gleave et al. (2020) allow different distributions to be used when computing $C^{\text{EPIC}}(R)$ and when taking the Pearson distance. However, doing this breaks some of their theoretical results. For details, see Appendix E.

**Definition 1.** A function $c : \mathcal{R} \rightarrow \mathcal{R}$ is a *canonicalisation function* if $c$ is linear, $c(R)$ and $R$ only differ by potential shaping and $S'$-redistribution for all $R \in \mathcal{R}$, and for all $R_1, R_2 \in \mathcal{R}$, $c(R_1) = c(R_2)$ if and only if $R_1$ and $R_2$ only differ by potential shaping and $S'$-redistribution.

Note that we require $c$ to be linear. Note also that $C^{\mathrm{EPIC}}$ and $C^{\mathrm{DARD}}$ are not canonicalisation functions in our sense, because we here require canonicalisation functions to simultaniously standardise both potential shaping and $S'$-redistribution, whereas $C^{\mathrm{EPIC}}$ and $C^{\mathrm{DARD}}$ only standardise potential shaping. In Section 2.2, we provide examples of canonicalisation functions. Let us next introduce the functions that we use to compute a distance:

**Definition 2.** A metric $m : \mathcal{R} \times \mathcal{R} \rightarrow \mathbb{R}$ is *admissible* if there exists a norm $p$ and two (positive) constants $u, \ell$ such that $\ell \cdot p(x, y) \leqslant m(x, y) \leqslant u \cdot p(x, y)$ for all $x, y \in \mathcal{R}$.

A metric is admissible if it is bilipschitz equivalent to a norm. Any norm is an admissible metric, though there are admissible metrics which are not norms.[3] Recall also that all norms are bilipschitz equivalent on any finite-dimensional vector space. This means that if $m$ satisfies Definition 2 for one norm, then it satisfies it for all norms. We can now define our class of reward metrics:

**Definition 3.** A function $d : \mathcal{R} \times \mathcal{R} \rightarrow \mathbb{R}$ is a *STARC metric* (STAndardised Reward Comparison) if there is a canonicalisation function $c$, a function $n$ that is a norm on $\mathrm{Im}(c)$, and a metric $m$ that is admissible on $\mathrm{Im}(s)$, such that $d(R_1, R_2) = m(s(R_1), s(R_2))$, where $s(R) = c(R)/n(c(R))$ when $n(c(R)) \neq 0$, and $c(R)$ otherwise.

Intuitively speaking, $c$ ensures that all reward functions which differ by potential shaping and $S'$-redistribution are considered to be equivalent, and division by $n$ ensures that positive scaling is ignored as well. Note that if $n(c(R)) = 0$, then $c(R)$ assigns 0 reward to every transition. Note also that $\mathrm{Im}(c)$ is the image of $c$, if $c$ is applied to the entirety of $\mathcal{R}$. If $n$ is a norm on $\mathcal{R}$, then $n$ is also a norm on $\mathrm{Im}(c)$, but there are functions which are norms on $\mathrm{Im}(c)$ but not on $\mathcal{R}$ (c.f. Proposition 4).

In Appendix C, we provide a geometric intuition for how STARC metrics work.

## 2.2 EXAMPLES OF STARC METRICS

In this section, we give several examples of STARC metrics. We begin by showing how to construct canonicalisation functions. We first give a simple and straightforward method:

**Proposition 2.** *For any policy $\pi$, the function $c : \mathcal{R} \rightarrow \mathcal{R}$ given by*

$$c(R)(s, a, s') = \mathbb{E}_{S' \sim \tau(s,a)} \left[ R(s, a, S') - V^\pi(s) + \gamma V^\pi(S') \right]$$

*is a canonicalisation function. Here $V^\pi$ is computed under the reward function $R$ given as input to $c$. We call this function Value-Adjusted Levelling (VAL).*

The proof, as well as all other proofs, are given in the Appendix. Proposition 2 gives us an easy way to make canonicalisation functions, which are also easy to evaluate whenever $V^\pi$ is easy to approximate. We next give another example of canonicalisation functions:

**Definition 4.** A canonicalisation function $c$ is *minimal* for a norm $n$ if for all $R$ we have that $n(c(R)) \leqslant n(R')$ for all $R'$ such that $R$ and $R'$ only differ by potential shaping and $S'$-redistribution.

Minimal canonicalisation functions give rise to tighter regret bounds (c.f. Section 3 and Appendix F). It is not a given that minimal canonicalisation functions exist for a given norm $n$, or that they are unique. However, for any weighted $L_2$-norm, this is the case:

**Proposition 3.** *For any weighted $L_2$-norm, a minimal canonicalisation function exists and is unique.*

A STARC metric can use any canonicalisation function $c$. Moreover, the normalisation step can use any function $n$ that is a norm on $\mathrm{Im}(c)$. This does of course include the $L_1$-norm, $L_2$-norm, $L_\infty$-norm, and so on. We next show that $\max_\pi J(\pi) - \min_\pi J(\pi)$ also is a norm on $\mathrm{Im}(c)$:

**Proposition 4.** *If $c$ is a canonicalisation function, then the function $n : \mathcal{R} \rightarrow \mathcal{R}$ given by $n(R) = \max_\pi J(\pi) - \min_\pi J(\pi)$ is a norm on $\mathrm{Im}(c)$.*

---

[3]For example, the unit ball of $m$ does not have to be convex, or symmetric around the origin.

For the final step we of course have that any norm is an admissible metric, though some other metrics are admissible as well.[4] To obtain a STARC metric, we then pick any canonicalisation function $c$, norm $n$, and admissible metric $m$, and combine them as described in Definition 3. Which choice of $c$, $n$, and $m$ is best in a given situation may depend on multiple considerations, such as how easy they are to compute, how easy they are to work with theoretically, or how well they together track worst-case regret (c.f. Section 3 and 4).

### 2.3 Unknown Transition Dynamics and Continuous Environments

STARC metrics depend on the transition function $\tau$, through the definition of canonicalisation functions (since $S'$-redistribution depends on $\tau$). Moreover, $\tau$ is often unknown in practice. However, it is important to note that while STARC metrics *depend* on $\tau$, there are STARC metrics that can be computed without *direct access* to $\tau$. For example, the VAL canonicalisation function (Proposition 2) only requires that we can *sample* from $\tau$, which is always possible in the reinforcement learning setting. Moreover, if we want to evaluate a learnt reward function in an environment that is different from the training environment, then we can simply use the $\tau$ from the evaluation environment. As such, we do not consider the dependence on $\tau$ to be a meaningful limitation. Nonetheless, it is possible to define STARC-like pseudometrics that do not depend on $\tau$ at all, and such pseudometrics also have some theoretical guarantees (albeit guarantees that are weaker than those enjoyed by STARC metrics). This option is discussed in Appendix F.3.

Moreover, we assume that $\mathcal{S}$ and $\mathcal{A}$ are finite, but many interesting environments are *continuous*. However, it is important to note that while our theoretical results assume that $\mathcal{S}$ and $\mathcal{A}$ are finite, it is still straightforward to compute and use STARC metrics in continuous environments (for example, using the VAL canonicalisation function from Proposition 2). We discuss this issue in more detail in Appendix D. In Section 4, we also provide experimental data from a continuous environment.

## 3 Theoretical Results

In this section, we prove that STARC metrics enjoy several desirable theoretical guarantees. First, we note that all STARC metrics are pseudometrics on the space of all reward functions, $\mathcal{R}$:

**Proposition 5.** *All STARC metrics are pseudometrics on $\mathcal{R}$.*

This means that STARC metrics give us a well-defined notion of a "distance" between rewards. Next, we characterise the cases when STARC metrics assign two rewards a distance of zero:

**Proposition 6.** *All STARC metrics have the property that $d(R_1, R_2) = 0$ if and only if $R_1$ and $R_2$ induce the same ordering of policies.*

This means that STARC metrics consider two reward functions to be equivalent, exactly when those reward functions induce exactly the same ordering of policies. This is intuitive and desirable.

For a pseudometric $d$ on $\mathcal{R}$ to be useful, it is crucial that it induces an upper bound on worst-case regret. Specifically, we want it to be the case that if $d(R_1, R_2)$ is small, then the impact of using $R_2$ instead of $R_1$ should also be small. When a pseudometric has this property, we say that it is *sound*:

**Definition 5.** A pseudometric $d$ on $\mathcal{R}$ is *sound* if there exists a positive constant $U$, such that for any reward functions $R_1$ and $R_2$, if two policies $\pi_1$ and $\pi_2$ satisfy that $J_2(\pi_2) \geqslant J_2(\pi_1)$, then

$$J_1(\pi_1) - J_1(\pi_2) \leqslant U \cdot (\max_\pi J_1(\pi) - \min_\pi J_1(\pi)) \cdot d(R_1, R_2).$$

Let us unpack this definition. $J_1(\pi_1) - J_1(\pi_2)$ is the regret, as measured by $R_1$, of using policy $\pi_2$ instead of $\pi_1$. Division by $\max_\pi J_1(\pi) - \min_\pi J_1(\pi)$ normalises this quantity based on the total range of $R_1$ (though the term is put on the right-hand side of the inequality, instead of being used as a denominator, in order to avoid division by zero when $\max_\pi J_1(\pi) - \min_\pi J_1(\pi) = 0$). The condition that $J_2(\pi_2) \geqslant J_2(\pi_1)$ says that $R_2$ prefers $\pi_2$ over $\pi_1$. Taken together, this means that a pseudometric $d$ on $\mathcal{R}$ is sound if $d(R_1, R_2)$ gives an upper bound on the maximal regret that could

---

[4]For example, if $m(x, y)$ is the *angle* between $x$ and $y$ when $x, y \neq 0$, and we define $m(0, 0) = 0$ and $m(x, 0) = \pi/2$ for $x \neq 0$, then $m$ is also admissible, even though $m$ is not a norm.

be incurred under $R_1$ if an arbitrary policy $\pi_1$ is optimised to another policy $\pi_2$ according to $R_2$. It is also worth noting that this includes the special case when $\pi_1$ is optimal under $R_1$ and $\pi_2$ is optimal under $R_2$. Our first main result is that all STARC metrics are sound:

**Theorem 1.** *All STARC metrics are sound.*

This means that any STARC metric gives us an upper bound on worst-case regret. Next, we will show that STARC metrics also induce a *lower* bound on worst-case regret. It may not be immediately obvious why this property is desirable. To see why this is the case, note that if a pseudometric $d$ on $\mathcal{R}$ does not induce a lower bound on worst-case regret, then there are reward functions that have a *low* worst-case regret, but a *large* distance under $d$. This would in turn mean that $d$ is not *tight*, and that it should be possible to improve upon it. In other words, if we want a small distance under $d$ to be both sufficient *and necessary* for low worst-case regret, then $d$ must induce both an upper *and a lower* bound on worst-case regret. As such, we also introduce the following definition:

**Definition 6.** A pseudometric $d$ on $\mathcal{R}$ is *complete* if there exists a positive constant $L$, such that for any reward functions $R_1$ and $R_2$, there exist two policies $\pi_1$ and $\pi_2$ such that $J_2(\pi_2) \geqslant J_2(\pi_1)$ and

$$J_1(\pi_1) - J_1(\pi_2) \geqslant L \cdot (\max_{\pi} J_1(\pi) - \min_{\pi} J_1(\pi)) \cdot d(R_1, R_2),$$

and moreover, if both $\max_{\pi} J_1(\pi) - \min_{\pi} J_1(\pi) = 0$ and $\max_{\pi} J_2(\pi) - \min_{\pi} J_2(\pi) = 0$, then we have that $d(R_1, R_2) = 0$.

The last condition is included to rule out certain pathological edge-cases. Intuitively, if $d$ is sound, then a small $d$ is *sufficient* for low regret, and if $d$ is complete, then a small $d$ is *necessary* for low regret. Soundness implies the absence of false positives, and completeness the absence of false negatives. Our second main result is that all STARC metrics are complete:

**Theorem 2.** *All STARC metrics are complete.*

Theorems 1 and 2 together imply that, for any STARC metric $d$, we have that a small value of $d$ is both necessary and sufficient for a low regret. This means that STARC metrics, in a certain sense, exactly capture what it means for two reward functions to be similar, and that we should not expect it to be possible to significantly improve upon them. We can make this claim formal as follows:

**Proposition 7.** *Any pseudometrics on $\mathcal{R}$ that are both sound and complete are bilipschitz equivalent.*

This implies that all STARC metrics are bilipschitz equivalent. Moreover, any other pseudometric on $\mathcal{R}$ that induces both an upper and a lower bound on worst-case regret (as we define it) must also be bilipschitz equivalent to STARC metrics.

In Appendix A and B, we provide an extensive analysis of both EPIC and DARD, and show that they fail to induce similar theoretical guarantees.

## 4 EXPERIMENTAL RESULTS

In this section we present our experimental results. First, we demonstrate that STARC metrics provide a better estimate of regret than EPIC and DARD in randomly generated MDPs. We then evaluate a STARC metric in a continuous environment.

### 4.1 LARGE NUMBERS OF SMALL RANDOM MDPS

Our first experiment compares several STARC metrics to EPIC, DARD, and a number of other non-STARC baselines. In total, our experiment covered 223 different pseudometrics (including rollout regret), derived by creating different combinations of canonicalisation functions, normalisations, and distance metrics. For details, see Appendix G.3. For each pseudometric, we generated a large number of random MDPs, and then measured how well the pseudometric correlates with *regret* across this distribution. The regret is defined analogously to Definition 5 and 6, except that only optimal policies are considered – for details, see Appendix G.2. We used MDPs with 32 states, 4 actions, $\gamma = 0.95$, a uniform initial state distribution, and randomly sampled sparse non-deterministic transition functions, and for each MDP, we generated several random reward functions. For details on the random generation process, see Appendix G. We compared 49,152 reward function pairs (Appendix G.4), and

used these to estimate how well each pseudometric correlates with regret. We show these correlations in Figure 1, and the full data is given in a table in Appendix H. In Appendix H.1, we also provide tables that indicate the impact of changing the metric $m$ or the normalisation function $n$.

The canonicalisation functions we used were `None` (which simply skips the canonicalisation step), $C^{\text{EPIC}}$, $C^{\text{DARD}}$, `MinimalPotential` (which is the minimal "canonicalisation" that removes potential shaping but not $S'$-redistribution, and therefore is easier to compute), `VALPotential` (which is given by $R(s, a, s') - V^{\pi}(s) + \gamma V^{\pi}(s')$), and `VAL` (defined in Proposition 2). For both $C^{\text{EPIC}}$ and $C^{\text{DARD}}$, both $\mathcal{D}_{\mathcal{S}}$ and $\mathcal{D}_{\mathcal{A}}$ were chosen to be uniform over $\mathcal{S}$ and $\mathcal{A}$. For both `VALPotential` and `VAL`, $\pi$ was chosen to be the uniformly random policy. Note that `VAL` is the only canonicalisation which removes both potential shaping and $S'$-redistribution, and thus the only one that meets Definition 1 — for this reason, it is listed as "STARC-VAL" in Figure 1. For the full details about which pseudometrics were chosen, and why, see Appendix G.3

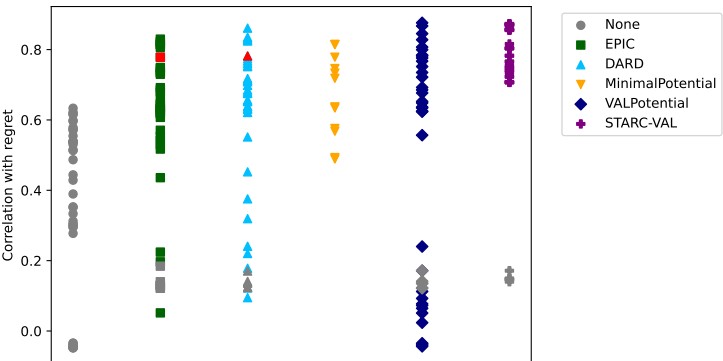

Figure 1: This figure displays the correlation to regret for several pseudometrics. Each point represents one pseudometric, i.e. one unique combination of canonicalisation $c$, normalisation $n$, and distance metric $m$. They are grouped together based on their canonicalisation function, with each column corresponding to a different canonicalisation function. Pseudometrics which skip canonicalisation or normalisation are shown in grey. The versions of EPIC and DARD that use the $L_2$ norm for both normalisation $n$ and distance metric $m$ are highlighted in red, as these are the original versions given in Gleave et al. (2020) and Wulfe et al. (2022). The STARC metrics, which are canonicalised using VAL, are reliably better indicators of regret than the other pseudometrics.

As we can see, the STARC metrics based on `VAL` perform noticeably better than all pre-existing pseudometrics – for instance, the correlation of EPIC to regret is 0.778, DARD's correlation is 0.782, while `VAL`'s correlation is 0.856 (when using $L_2$ for both $n$ and $m$, which is the same as EPIC and DARD). Out of the 10 best pseudometrics, 8 use `VAL` (and the other 2 both use `VALPotential`). Moreover, for each choice of $n$ and $m$, we have that the `VAL` canonicalisation performs better than the EPIC canonicalisation in 40 out of 42 cases.[5] Taken together, these results suggest that STARC metrics robustly perform better than the existing alternatives.

Our results also suggest that the choice of normalisation function $n$ and metric $m$ can have a significant impact on the pseudometric's accuracy. For instance, when canonicalising with `VAL`, it is better to use the $L_1$ norm than the $L_2$ norm for both normalisation and taking the distance – this increases the correlation with regret from 0.856 to 0.873. Another example is the EPIC canonicalisation – when paired with the weighted $L_{\infty}$ norm for normalisation and the (unweighted) $L_{\infty}$ norm for taking the distance, instead of using the $L_2$ norm for both, its correlation decreases from 0.778 to 0.052. As we can see in Figure 1, this effect appears to be more prominent for the non-STARC metrics. Another thing to note is that it seems like `VALPotential` can perform as well as `VAL` despite not canonicalising for $S'$-redistribution, but only when a ($\tau$-)weighted norm is used. This may be because $\tau$-weighted norms set all impossible transitions to 0, and reduce the impact of very unlikely transitions; plausibly, this could in practice be similar to canonicalising for $S'$-redistribution. When using `VAL`, $L_1$ was the best unweighted norm for both $m$ and $n$ in our experiment.

---

[5]The only exceptions are when no normalisation is used and $m = L_{\infty}$, and when $n = \text{weighted-}L_2$ and $m = \text{weighted-}L_{\infty}$. However, in the first case, both the EPIC-based and the VAL-based pseudometric perform badly (since no normalisation is used), and in the second case, the difference between them is not large.

### 4.2 THE REACHER ENVIRONMENT

Our next experiment estimates the distance between several hand-crafted reward functions in the Reacher environment from MuJoCo (Todorov et al., 2012). This is a deterministic environment with an 11-dimensional continuous state space and a 2-dimensional continuous action space. The reward functions we used are:

1. `GroundTruth`: The Euclidean distance to the target, plus a penalty term for large actions.
2. `PotentialShaped`: `GroundTruth` with random potential shaping.
3. `SecondPeak`: We create a second target in the environment, and reward the agent based on both its distance to this target, and to the original target, but give a greater weight to the original target.
4. `Random`: A randomly generated reward, implemented as an affine transformation from $s, a, s'$ to real numbers with the weights and bias randomly initialised.
5. `Negative`: Returns $-$`GroundTruth`.

We expect `GroundTruth` to be equivalent to `PotentialShaped`, similar to `SecondPeak`, orthogonal to `Random`, and opposite to `Negative`. We used the VAL canonicalisation function with the uniform policy, and normalised and took the distance with the $L_2$-norm. This pseudometric was then estimated through sampling; full details can be found in Appendix D and I. The results of this experiment are given in Table 1. As we can see, the relative ordering of the reward functions match what we expect. However, the magnitudes of the estimated distances are noticeably larger than their real values; for example, the actual distance between `GroundTruth` and `PotentialShaped` is 0, but it is estimated as $\approx 0.9$. The reason for this is likely that the estimation involves summing over absolute values, which makes all noise positive. Nonetheless, for the purposes of *ranking* the rewards, this is not fundamentally problematic.

| PotentialShaped | SecondPeak | Random | Negative |
|---|---|---|---|
| 0.8968 | 1.2570 | 1.3778 | 1.706 |

Table 1: This figure displays the estimated distance (using $c = $ VAL, $n = L_2$, and $m = L_2$) between each reward function in the Reacher environment and the `GroundTruth` reward function.

## 5 DISCUSSION

We have introduced STARC metrics, and demonstrated that they provide compelling theoretical guarantees. In particular, we have shown that they are both sound and complete, which means that they induce both an upper and a lower bound on worst-case regret. As such, a small STARC distance is both necessary and sufficient to ensure that two reward functions induce a similar ordering of policies. Moreover, any two pseudometrics that are both sound and complete must be bilipschitz equivalent. This means that any pseudometric on $\mathcal{R}$ that has the same theoretical guarantees as STARC metrics must be equivalent to STARC metrics. This means that we have provided what is essentially a complete answer to the question of how to correctly measure the distance between reward functions. Moreover, our experiments show that STARC metrics have a noticeably better empirical performance than any existing pseudometric in the current literature, for a wide range of environments. This means that STARC metrics offer direct practical advantages, in addition to their theoretical guarantees. In addition to this, STARC metrics are both easy to compute, and easy to work with mathematically. As such, STARC metrics will be useful for both empirical and theoretical work on the analysis and evaluation of reward learning algorithms.

Our work can be extended in a number of ways. First of all, it would be desirable to establish more conclusively which STARC metrics work best in practice. Our experiments are indicative, but not conclusive. Secondly, our theoretical results assume that $\mathcal{S}$ and $\mathcal{A}$ are finite; it would be desirable to generalise them to continuous environments. Third, we use a fairly strong definition of regret. We could consider some weaker criterion, that may allow for the creation of more permissive reward metrics. Finally, our work considers the MDP setting – it would be interesting to also consider other classes of environments. We believe that the multi-agent setting would be of particular interest, since it introduces new and more complex dynamics that are not present in the case of MDPs.

ACKNOWLEDGEMENTS

The authors wish to acknowledge and thank the financial support of the UK Research and Innovation (UKRI) [Grant ref EP/S022937/1] and the University of Bristol.

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

## A  COMMENTS ON EPIC

In this appendix, we will show that EPIC has a number of undesirable properties. For the sake of readability, the proofs of the theorems in this section are given in Appendix F.3, rather than here.

First of all, while EPIC does induce an upper bound on a form of regret, this is not the type of regret that is typically relevant in practice. To demonstrate this, we will first provide a generalisation of the regret bound given in Gleave et al. (2020):

**Theorem 3.** *There exists a positive constant $U$, such that for any reward functions $R_1$ and $R_2$, and any $\tau$ and $\mu_0$, if two policies $\pi_1$ and $\pi_2$ satisfy that $J_2(\pi_2) \geqslant J_2(\pi_1)$, then we have that*

$$J_1(\pi_1) - J_1(\pi_2) \leqslant U \cdot L_2(R_1) \cdot D^{\mathrm{EPIC}}(R_1, R_2).$$

Theorem 3 covers the special case when $\pi_1$ is optimal under $R_1$, and $\pi_2$ is optimal under $R_2$. This means that this theorem is a generalisation of the bound given in Gleave et al. (2020). However, we will next show that EPIC does not induce a regret bound of the form given in Definition 5:

**Theorem 4.** *EPIC is not sound, for any choice of $\tau$ or $\mu_0$.*

These results may at first seem paradoxical, since the definition of soundness is quite similar to the statement of Theorem 3. The reason why these statements can both be true simultaneously is that $L_2(R_1)$ can be arbitrarily large even as $\max_\pi J_1(\pi) - \min_\pi J_1(\pi)$ becomes arbitrarily small. Moreover, we argue that it is more relevant to compare $J_1(\pi_1) - J_1(\pi_2)$ to $\max_\pi J_1(\pi) - \min_\pi J_1(\pi)$, rather than $L_2(R_1)$. First of all, note that it is not informative to just know the absolute value of $J_1(\pi_1) - J_1(\pi_2)$, since this depends on the scale of $R_1$. Rather, what we want to know how much reward we might lose, relative to the total amount of reward that could be had. This quantity is measured by $\max_\pi J_1(\pi) - \min_\pi J_1(\pi)$, not $L_2(R_1)$. For example, if $J_1(\pi_1) - J_1(\pi_2) = \epsilon$, but $\max_\pi J_1(\pi) - \min_\pi J_1(\pi)$ is just barely larger than $\epsilon$, then this should be considered to be a large loss of reward, even if $L_2(R_1) \gg \epsilon$. For this reason, we consider the regret bound given in Definition 5 to be more informative than the regret bound given in Gleave et al. (2020) and in Theorem 3.

Another relevant question is whether EPIC induces a *lower* bound on worst-case regret. We next show that this is not the case, regardless of whether we consider the type of regret used in Gleave et al. (2020), or the type of regret used in Theorem 3, or the type of regret used in Definition 5:

**Theorem 5.** *There exist rewards $R_1$, $R_2$ such that $D^{\mathrm{EPIC}}(R_1, R_2) > 0$, but where $R_1$ and $R_2$ induce the same ordering of policies for any choice of $\tau$ or $\mu_0$.*

This means that EPIC cannot induce a lower bound on worst-case regret, for almost any way of defining regret. Together, we think these results show that EPIC lacks the theoretical guarantees that we desire in a reward function pseudometric.

## B  COMMENTS ON DARD

In this appendix, we briefly discuss some of the properties of DARD. In so doing, we will criticise some of the choices made in the design of DARD, and argue that STARC metrics offer a better way to incorporate information about the environment dynamics.

To start with, recall that DARD uses the canonicalisation function $C^{\mathrm{DARD}}$, which is the function where $C^{\mathrm{DARD}}(R)(s, a, s')$ is given by

$$R(s, a, s') + \mathbb{E}[\gamma R(s', A, S'') - R(s, A, S') - \gamma R(S', A, S'')],$$

where $A \sim \mathcal{D}_\mathcal{A}$, $S' \sim \tau(s, A)$, and $S'' \sim \tau(s', A)$. Moreover, also recall that $C^{\mathrm{DARD}}$ only is designed to remove potential shaping, whereas the canonicalisation functions we specify in Definition 1 are designed to remove both potential shaping and $S'$-redistribution.

Now, first and foremost, note that while $C^{\mathrm{DARD}}$ is designed to remove potential shaping, it does this in a somewhat strange way. In particular, while it is shown in Wulfe et al. (2022) that $C^{\mathrm{DARD}}(R_1) = C^{\mathrm{DARD}}(R_2)$ if $R_1$ and $R_2$ differ by potential shaping, it is in general *not* the case that $R$ and $C^{\mathrm{DARD}}(R)$ differ by potential shaping. To see this, note that the term $\gamma R(S', A, S'')$ depends on both $s$ and $s'$, which a potential shaping function cannot do. This has a few important consequences.

In particular, it is unclear if $C^{\mathrm{DARD}}(R_1) = C^{\mathrm{DARD}}(R_2)$ *only if* $R_1$ and $R_2$ differ by potential shaping. It is also unclear if $R$ and $C^{\mathrm{DARD}}(R)$ in general even have the same policy ordering. Note also that $C^{\mathrm{DARD}}(R)$ is not monotonic, in the sense that $C^{\mathrm{DARD}}(C^{\mathrm{DARD}}(R))$ and $C^{\mathrm{DARD}}(R)$ may be different. This seems undesirable.

Another thing to note is that $\mathbb{E}[C^{\mathrm{DARD}}(R)(S, A, S')]$ may not be 0. This means that it is unclear whether or not DARD can be expressed in terms of norms, like EPIC can (c.f. Proposition 12). It is also unclear if DARD induces an upper bound on regret, since Wulfe et al. (2022) do not provide a regret bound. The fact that $C^{\mathrm{DARD}}(R)$ is not a potential shaping function does not necessarily imply that DARD does not induce an upper bound on regret. However, without a proof, there is a worry that there might be reward pairs with a bounded DARD distance but unbounded regret.

Yet another thing to note is that, while DARD is designed to be used in cases where the environment dynamics are known, it can still be influenced by the reward of transitions that are impossible according to the environment dynamics. For example, the final term of $C^{\mathrm{DARD}}(R)(s, a, s')$ can be influenced by impossible transitions. This gives us the following result:

**Proposition 8.** *There exists transition functions $\tau$ and initial state distributions $\mu_0$ for which DARD is not complete.*

*Proof.* Consider an environment $(\mathcal{S}, \mathcal{A}, \tau, \mu_0, \_, \gamma)$ where $\mathcal{S} = \{s_1, s_2, s_3\}$, $\mathcal{A} = \{a_1, a_2\}$, and where the transition function is given by $\tau(s_1, a) = s_2$, $\tau(s_2, a) = s_3$, and $\tau(s_3, a) = s_1$, for any $a \in \mathcal{A}$. We may also suppose $\mu_0 = s_1$, and $\gamma = 0.9$.

Next, let $R_1 = R_2 = 0$ for all transitions which are possible under $\tau$, but let $R_1(s, a_1, s') = 1$, $R_1(s, a_2, s') = 0$, $R_2(s, a_1, s') = 0$, and $R_2(s, a_2, s') = 1$ for all transitions which are impossible under $\tau$. Now $D^{\mathrm{DARD}}(R_1, R_2) > 0$, even though $(\mathcal{S}, \mathcal{A}, \tau, \mu_0, R_1, \gamma)$ and $(\mathcal{S}, \mathcal{A}, \tau, \mu_0, R_2, \gamma)$ have exactly the same policy ordering. $\square$

Together, the above leads us to worry that DARD might lead to misleading measurements. Therefore, we believe that STARC metrics offer a better way to incorporate knowledge about the transition dynamics into the reward metric, especially in the light of our results from Section 3.

## C   A Geometric Intuition for STARC Metrics

In this section, we will provide a geometric intuition for how STARC metrics work. This will help to explain why STARC metrics are designed in the way that they are, and how they work. It may also make it easier to understand some of our proofs.

First of all, note that the space of all reward functions $\mathcal{R}$ forms an $|\mathcal{S}||\mathcal{A}||\mathcal{S}|$-dimensional vector space. Next, recall that if two reward functions $R_1$ and $R_2$ differ by (some combination of) potential shaping and $S'$-redistribution, then $R_1$ and $R_2$ induce the same ordering of policies. Moreover, both of these transformations are *additive*. In other words, they correspond to a set of reward functions $\{R_0\}$, such that $R_1$ and $R_2$ differ by a combination of potential shaping and $S'$-redistribution if and only if $R_1 - R_2 \in \{R_0\}$. This means that $\{R_0\}$ is a linear subspace of $\mathcal{R}$, and that for any reward function $R$, the set of all reward functions that differ from $R$ by a combination of potential shaping and $S'$-redistribution together form an affine subspace of $\mathcal{R}$.

A canonicalisation function is a linear map that removes the dimensions that are associated with $\{R_0\}$. In other words, they map $\mathcal{R}$ to an $|\mathcal{S}|(|\mathcal{A}| - 1)$-dimensional subspace of $\mathcal{R}$ in which no reward functions differ by potential shaping or $S'$-redistribution. The null space of a canonicalisation function is always $\{R_0\}$. The canonicalisation function that is minimal for the $L_2$-norm is the orthogonal map that satisfies these properties, whereas other canonicalisation functions are non-orthogonal.

When we normalise the resulting reward functions by dividing by a norm $n$, we project the entire vector space onto the unit ball of $n$ (except the zero reward, which remains at the origin). The metric $m$ then measures the distance between the resulting reward functions on the surface of this sphere:

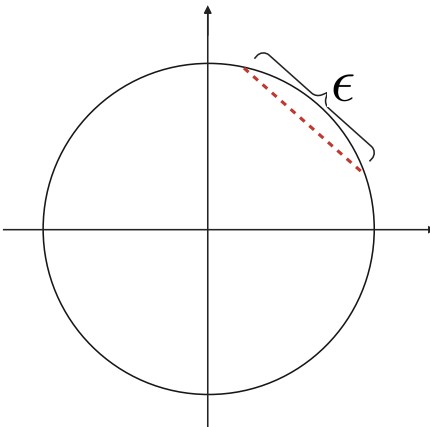

To make this more clear, it may be worth considering the case of non-sequential decision making. Suppose we have a finite set of *choices* $C$, and a utility function $U : C \to \mathbb{R}$. Given two distributions $\mathcal{D}_1$, $\mathcal{D}_2$ over $C$, we say that we prefer $\mathcal{D}_1$ over $\mathcal{D}_2$ if $\mathbb{E}_{c \sim \mathcal{D}_1}[U(c)] > \mathbb{E}_{c \sim \mathcal{D}_2}[U(c)]$. The set of all utility functions over $C$ forms a $|C|$-dimensional vector space. Moreover, in this setting, it is well-known that two utility functions $U_1$, $U_2$ induce the same preferences between all possible distributions over $C$ if and only if they differ by an affine transformation. Therefore, if we wanted to represent the set of all *non-equivalent* utility functions over $C$, we may consider requiring that $U(c_0) = 0$ for some $c_0 \in C$, and that $L_2(U) = 1$ unless $U(c) = 0$ for all $c \in C$. Any utility function over $C$ is equivalent to some utility function in this set, and this set can in turn be represented as the surface of a $(|C| - 1)$-dimensional sphere, together with the origin.

This is essentially analogous to the normalisation that the canonicalisation function $c$ and the normalisation function $n$ perform for STARC metrics. Here $C$ is analogous to the set of all trajectories, the trajectory return function $G$ is analogous to $U$, and a policy $\pi$ induces a distribution over trajectories. It is worth knowing that affine transformations of the trajectory return function, $G$, correspond exactly to potential shaping and positive linear scaling of $R$ (see Skalse et al., 2022a, their Theorem 3.12). However, while the cases are analogous, it is not a direct correspondence, because not all distributions over trajectories can be realised as a policy in a given MDP.

Another perspective that may help with understanding STARC metrics comes from considering *occupancy measures*. Specifically, for a given policy $\pi$, let its occupancy measure $\eta^\pi$ be the $|\mathcal{S}||\mathcal{A}||\mathcal{S}|$-dimensional vector in which the value of the $(s, a, s')$'th dimension is

$$\sum_{t=0}^{\infty} \gamma^t \mathbb{P}_{\xi \sim \pi}(S_t = s, A_t = a, S_{t+1} = s').$$

Now note that $J(\pi) = \eta^\pi \cdot R$. Therefore, by computing occupancy measures, we can divide the computation of $J$ into two parts, the first of which is independent of $R$, and the second of which is a linear function. Moreover, let $\Omega = \{\eta^\pi : \pi \in \Pi\}$ be the set of all occupancy measures. We now have that the policy value function $J$ of a reward function $R$ can be visualised as a linear function on this set. Moreover, if we have two reward functions $R_1$, $R_2$, then they can be visualised as two different linear functions on this set:

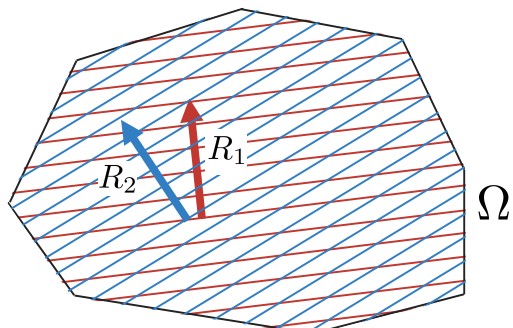

$\Omega$ is located in a $|\mathcal{S}|(|\mathcal{A}| - 1)$-dimensional affine subspace of $\mathbb{R}^{|\mathcal{S}||\mathcal{A}||\mathcal{S}|}$, and contains a set which is open in this space (Skalse & Abate, 2023). Moreover, it can be represented as the convex hull of a finite set of points (Feinberg & Rothblum, 2012). It is thus a polytope.

From this image, it is visually clear that the worst-case regret of maximising $R_1$ instead of $R_2$, should be proportional to the angle between the projections of $R_1$ and $R_2$ onto $\Omega$. Moreover, this is what STARC metrics measure; any STARC metric is bilipschitz equivalent to the angle between reward functions projected onto $\Omega$. This in should in turn give an intuition for why the STARC distance between two rewards provide both an upper and lower bound on their worst-case regret.

## D    APPROXIMATING STARC METRICS IN LARGE ENVIRONMENTS

In small MDPs, STARC metrics can be computed exactly (in time that is polynomial in $|\mathcal{S}|$ and $|\mathcal{A}|$). However, most realistic MDPs are too large for this to be feasible. As such, we will here discuss how to approximate STARC metrics in large environments, including continuous environments.

First, recall the VAL canonicalisation function (Proposition 2), given by

$$c(R)(s, a, s') = \mathbb{E}_{S' \sim \tau(s,a)} \left[ R(s, a, S') - V^\pi(s) + \gamma V^\pi(S') \right],$$

where $\pi$ can be any (fixed) policy. This canonicalisation function is straightforward to approximate in any environment where reinforcement learning can be used, including large-scale environments. To do this, first pick an arbitrary policy $\pi$, such as e.g. the uniformly random policy. Then compute an approximation of $V^\pi$ – this can be done using a neural network updated with *on-policy* Bellman updates (Sutton & Barto, 2018). Note that $\pi$ should not be updated, and must remain fixed. Then simply estimate the expected value of $R(s, a, S') - V^\pi(s) + \gamma V^\pi(S')$ by sampling from $\tau$. This approximation can be computed in any environment where it is possible to sample from $\tau$ and approximate $V^\pi$ (which is to say, any environment where reinforcement learning is applicable). Note that we do not require direct access to $\tau$, we only need to be able to sample from it.

Next, note that if $v$ is an $n$-dimensional vector, and $\mathcal{U}$ is the uniform distribution over $\{1 \ldots n\}$, then

$$L_p(v) = (n \cdot \mathbb{E}_{i \sim \mathcal{U}}[|v_i|^p])^{1/p} = n^{1/p} \cdot \mathbb{E}_{i \sim \mathcal{U}}[|v_i|^p]^{1/p}.$$

This in turn means that

$$L_p\left(\frac{v}{L_p(v)}, \frac{w}{L_p(w)}\right) = \mathbb{E}_{i \sim \mathcal{U}}\left[\left|\frac{v_i}{\mathbb{E}_{j \sim \mathcal{U}}[|v_j|^p]^{1/p}} - \frac{w_i}{\mathbb{E}_{j \sim \mathcal{U}}[|w_j|^p]^{1/p}}\right|^p\right]^{1/p}$$

since the $n^{1/p}$-terms cancel out. Therefore, the normalisation step and distance step can also be estimated through sampling; simply sample enough random transitions to approximate each of the expectations in the expression above. Indeed, this can even be done if the state space and action space are continuous. Recall that the $L_p$-norm of an infinite-dimensional vector $v$ is defined as

$$L_{p,X}(v) = \left(\int_X |v_x|^p \mathrm{d}x\right)^{1/p}$$

where $X \subseteq \mathbb{R}^n$. This value can also be approximated through sampling.

It is also possible to approximate division by $n$ and taking the distance with $m$ using the Pearson distance, which is what EPIC does. In particular, let $\mathcal{D}$ be a distribution over $\mathcal{S} \times \mathcal{A} \times \mathcal{S}$ that assigns positive probability to all transitions, and let $R_1$, $R_2$ be reward functions such that $\mathbb{E}_{S,A,S' \sim \mathcal{D}}[R_1(S, A, S')] = \mathbb{E}_{S,A,S' \sim \mathcal{D}}[R_1(S, A, S')] = 0$. We then have that the Pearson distance $\sqrt{(1 - \rho(R_1(S, A, S'), R_2(S, A, S')))/2}$ between $R_1(S, A, S')$ and $R_2(S, A, S')$, where $S, A, S' \sim \mathcal{D}$, is equal to

$$\frac{1}{2} \cdot L_{2,W} \left( \frac{R_1}{L_{2,W}(R_1)}, \frac{R_2}{L_{2,W}(R_2)} \right),$$

where $W$ is a weight matrix depending on $\mathcal{D}$, and $\rho$ denotes the Pearson correlation. For details, see the proof of Proposition 12. For this identity to hold, it is crucial that $\mathbb{E}_{S,A,S' \sim \mathcal{D}}[R_1(S, A, S')] = \mathbb{E}_{S,A,S' \sim \mathcal{D}}[R_1(S, A, S')] = 0$. However, this can easily be ensured; for an arbitrary canonicalisation function $c_1$, let $c_2(R) = c_1(R) - \mathbb{E}_{S,A,S' \sim \mathcal{D}}[c_1(R)(S, A, S')]$. If $c_1$ is a valid canonicalisation function, then so is $c_2$. The Pearson correlation can of course be estimated through sampling.

# E    PROOFS OF MISCELLANEOUS CLAIMS

In this Appendix, we provide proofs for several miscellaneous claims and minor propositions made throughout the paper, especially in Section 2.2.

**Proposition 9.** *For any policy $\pi$, the function $c : \mathcal{R} \to \mathcal{R}$ given by*

$$c(R)(s, a, s') = \mathbb{E}_{S' \sim \tau(s,a)} \left[ R(s, a, S') - V^\pi(s) + \gamma V^\pi(S') \right]$$

*is a canonicalisation function.*

*Proof.* To prove that $c$ is a canonicalisation function, we must show

1. that $c$ is linear,

2. that $c(R)$ and $R$ only differ by potential shaping and $S'$-redistribution, and

3. that $c(R_1) = c(R_2)$ if and only if $R_1$ and $R_2$ only differ by potential shaping and $S'$-redistribution.

We first show that $c$ is linear. Given a state $s$, let $v_s$ be the $|\mathcal{S}||\mathcal{A}||\mathcal{S}|$-dimensional vector where

$$v_s[s', a, s''] = \sum_{i=0}^{\infty} \gamma^i \cdot \mathbb{P}(S_i = s', A_i = a, S_{i+1} = s''),$$

where the probability is given for a trajectory that is generated from $\pi$ and $\tau$, starting in $s$. Now note that $V^\pi(s) = v_s \cdot R$, where $R$ is represented as a vector. Using these vectors $\{v_s\}$, it is possible to express $c$ as a linear transformation.

To see that $c(R)$ and $R$ differ by potential shaping and $S'$-redistribution, it is sufficient to note that $V^\pi$ acts as a potential function, and that setting $R_2(s, a, s') = \mathbb{E}_{S' \sim \tau(s,a)}[R_1(s, a, S')]$ is a form of $S'$-redistribution.

To see that $c(R_1) = c(R_2)$ if $R_1$ and $R_2$ differ by potential shaping and $S'$-redistribution, first note that if $R_1$ and $R_2$ differ by potential shaping, so that $R_2(s, a, s') = R_1(s, a, s') + \gamma \Phi(s') - \Phi(s)$ for some $\Phi$, then $V_2^\pi(s) = V_1^\pi(s) - \Phi(s)$ (see e.g. Lemma B.1 in Skalse et al., 2022a). This means that

$$\begin{aligned}
c(R_2)(s, a, s') &= \mathbb{E}[R_2(s, a, S') + \gamma \cdot V_2^\pi(S') - V_2^\pi(s)] \\
&= \mathbb{E}[R_1(s, a, S') + \gamma \cdot \Phi(S') - \Phi(s) + \gamma \cdot (V_1^\pi(S') - \Phi(S')) - (V_1^\pi(s) - \Phi(s))] \\
&= \mathbb{E}[R_1(s, a, S') + \gamma \cdot V_1^\pi(S') - V_1^\pi(s)] \\
&= c(R_1)(s, a, s').
\end{aligned}$$

To see that $c(R_1) = c(R_2)$ only if $R_1$ and $R_2$ differ by potential shaping and $S'$-redistribution, first note that we have already shown that $R$ and $c(R)$ differ by potential shaping and $S'$-redistribution for all $R$. This implies that $R_1$ and $c(R_1)$ differ by potential shaping and $S'$-redistribution, and likewise for $R_2$ and $c(R_2)$. Then if $c(R_1) = c(R_2)$, we can combine these transformations, and obtain that $R_1$ and $R_2$ also differ by potential shaping and $S'$-redistribution. This completes the proof. $\qquad \square$

Note that this holds regardless of which environment $V^\pi$ is calculated in. Therefore, $V^\pi$ could be computed in an environment that is entirely distinct from the training environment. For example, $V^\pi$ could be computed in an environment (and using a policy) designed specifically to make this computation easy.

**Proposition 10.** *For any weighted $L_2$-norm, a minimal canonicalisation function exists and is unique.*

*Proof.* Let $R_0$ be the reward function that is 0 for all transitions. First note that the set of all reward functions that differ from $R_0$ by potential shaping and $S'$-redistribution form a linear subspace of $\mathcal{R}$. Let this space be denoted by $\mathcal{Y}$, and let $\mathcal{X}$ denote the orthogonal complement of $\mathcal{Y}$ in $\mathcal{R}$. Now any reward function $R \in \mathcal{R}$ can be uniquely expressed in the form $R_\mathcal{X} + R_\mathcal{Y}$, where $R_\mathcal{X} \in \mathcal{X}$ and $R_\mathcal{Y} \in \mathcal{Y}$. Consider the function $c : \mathcal{R} \to \mathcal{R}$ where $c(R) = R_\mathcal{X}$. Now this function is a canonicalisation function such that $n(c(R)) \leqslant R'$ for all $R'$ such that $c(R) = c(R')$, assuming that $n$ is a weighted $L_2$-norm.

To see this, we must show that

1. $c$ is linear,

2. $c(R)$ and $R$ differ by potential shaping and $S'$-redistribution,

3. $c(R_1) = c(R_2)$ if $R_1$ and $R_2$ differ by potential shaping and $S'$-redistribution, and

4. $n(c(R)) \leqslant n(R')$ for all $R'$ such that $c(R) = c(R')$.

It follows directly from the construction that $c$ is linear. To see that $c(R)$ and $R$ differ by potential shaping and $S'$-redistribution, simply note that $c(R) = R - R_\mathcal{Y}$, where $R_\mathcal{Y}$ is given by a combination of potential shaping and $S'$-redistribution of $R_0$. To see that $c(R_1) = c(R_2)$ if $R_1$ and $R_2$ differ by potential shaping and $S'$-redistribution, let $R_2 = R_1 + R'$, where $R'$ is given by potential shaping and $S'$-redistribution of $R_0$, and let $R_1 = R_\mathcal{X} + R_\mathcal{Y}$, where $R_\mathcal{X} \in \mathcal{X}$ and $R_\mathcal{Y} \in \mathcal{Y}$. Now $c(R_1) = R_\mathcal{X}$. Moreover, $R_2 = R_\mathcal{X} + R_\mathcal{Y} + R'$. We also have that $R' \in \mathcal{Y}$. We can thus express $R_2$ as $R_\mathcal{X} + (R_\mathcal{Y} + R')$, where $R_\mathcal{X} \in \mathcal{X}$ and $(R_\mathcal{Y} + R') \in \mathcal{Y}$, which implies that $c(R_2) = R_\mathcal{X}$. Therefore, if $R_1$ and $R_2$ differ by potential shaping and $S'$-redistribution, then $c(R_1) = c(R_2)$. To see that $c(R_1) = c(R_2)$ only if $R_1$ and $R_2$ differ by potential shaping and $S'$-redistribution, first note that we have already shown that $R$ and $c(R)$ differ by potential shaping and $S'$-redistribution for all $R$. This implies that $R_1$ and $c(R_1)$ differ by potential shaping and $S'$-redistribution, and likewise for $R_2$ and $c(R_2)$. Then if $c(R_1) = c(R_2)$, we can combine these transformations, and obtain that $R_1$ and $R_2$ also differ by potential shaping and $S'$-redistribution.

To see that $n(c(R)) \leqslant n(R')$ for all $R'$ such that $c(R) = c(R')$, first note that if $c(R) = c(R')$, then $R = R_\mathcal{X} + R_\mathcal{Y}$ and $R' = R_\mathcal{X} + R'_\mathcal{Y}$, where $R_\mathcal{X} \in \mathcal{X}$ and $R_\mathcal{Y}, R'_\mathcal{Y} \in \mathcal{Y}$. This means that $n(c(R)) = n(R_\mathcal{X})$, and $n(R') = n(R_\mathcal{X} + R'_\mathcal{Y})$. Moreover, since $n$ is a weighted $L_2$-norm, and since $R_\mathcal{X}$ and $R'_\mathcal{Y}$ are orthogonal, we have that $n(R_\mathcal{X} + R_\mathcal{Y}) = \sqrt{n(R_\mathcal{X})^2 + n(R_\mathcal{Y})^2} \geqslant n(R_\mathcal{X})$. This means that $n(c(R)) \leqslant n(R')$.

To see that this canonicalisation function is the unique minimal canonicalisation function for any weighted $L_2$-norm $n$, consider an arbitrary reward function $R$. Now, the set of all reward functions that differ from $R$ by potential shaping and $S'$-redistribution forms an affine space of $\mathcal{R}$, and a minimal canonicalisation function must map $R$ to a point $R'$ in this space such that $n(R') \leqslant n(R'')$ for all other points $R''$ in that space. If $n$ is a weighted $L_2$-norm, then this specifies a convex optimisation problem with a unique solution. $\qquad\square$

Note that this proof only shows that a minimal canonicalisation function exists and is unique when $n$ is a (weighted) $L_2$-norm. It does not show that such a canonicalisation function *only* exists for these norms, nor does it show that it is unique for all norms.

**Proposition 11.** *If $c$ is a canonicalisation function, then the function $n : \mathcal{R} \to \mathcal{R}$ given by $n(R) = \max_\pi J(\pi) - \min_\pi J(\pi)$ is a norm on $\mathrm{Im}(c)$.*

*Proof.* To show that a function $n$ is a norm on $\mathrm{Im}(c)$, we must show that it satisfies:

1. $n(R) \geqslant 0$ for all $R \in \text{Im}(c)$.

2. $n(R) = 0$ if and only if $R = R_0$ for all $R \in \text{Im}(c)$.

3. $n(\alpha \cdot R) = \alpha \cdot n(R)$ for all $R \in \text{Im}(c)$ and all scalars $\alpha$.

4. $n(R_1 + R_2) \leqslant n(R_1) + n(R_2)$ for all $R_1, R_2 \in \text{Im}(c)$.

Here $R_0$ is the reward function that is 0 everywhere. It is trivial to show that Axioms 1 and 3 are satisfied by $n$. For Axiom 2, note that $n(R) = 0$ exactly when $\max_\pi J(\pi) = \min_\pi J(\pi)$. If $R$ is $R_0$, then $J(\pi) = 0$ for all $\pi$, and so the "if" part holds straightforwardly. For the "only if" part, let $R$ be a reward function such that $\max_\pi J(\pi) = \min_\pi J(\pi)$. Then $R$ and $R_0$ induce the same policy ordering under $\tau$ and $\mu_0$, which means that they differ by potential shaping, $S'$-redistribution, and positive linear scaling (see Proposition 1). Moreover, since $R_0$ is 0 everywhere, this means that $R$ and $R_0$ in fact differ by potential shaping and $S'$-redistribution. However, from the definition of canonicalisation functions, if $R_1, R_2 \in \text{Im}(c)$ differ by potential shaping and $S'$-redistribution, then it must be that $R_1 = R_2$. Hence Axiom 2 holds as well.

We can show that Axiom 4 holds algebraically:

$$
\begin{aligned}
n(R_1 + R_2) &= \max_\pi (J_1(\pi) + J_2(\pi)) - \min_\pi (J_1(\pi) + J_2(\pi)) \\
&\leqslant \max_\pi J_1(\pi) + \max_\pi J_2(\pi) - \min_\pi J_1(\pi) - \min_\pi J_2(\pi) \\
&= (\max_\pi J_1(\pi) - \min_\pi J_1(\pi)) + (\max_\pi J_2(\pi) - \min_\pi J_2(\pi)) \\
&= n(R_1) + n(R_2)
\end{aligned}
$$

This means that $n(R) = \max_\pi J(\pi) - \min_\pi J(\pi)$ is a norm on $\text{Im}(c)$. $\qquad\square$

This means that we can normalise the reward functions so that $\max_\pi J(\pi) - \min_\pi J(\pi) = 1$, which is nice. This proposition will also be useful in our later proofs.

**Proposition 12.** *EPIC can be expressed as*

$$
D^{\text{EPIC}}(R_1, R_2) = \frac{1}{2} \cdot L_{2,\mathcal{D}} \left( \frac{C^{\text{EPIC}}(R_1)}{L_{2,\mathcal{D}}(C^{\text{EPIC}}(R_1))}, \frac{C^{\text{EPIC}}(R_2)}{L_{2,\mathcal{D}}(C^{\text{EPIC}}(R_2))} \right),
$$

*where $L_{2,\mathcal{D}}$ is a weighted $L_2$-norm.*

*Proof.* Recall that by default, $D^{\text{EPIC}}(R_1, R_2)$ is defined to be the Pearson distance between $C^{\text{EPIC}}(R_1)(S, A, S')$ and $C^{\text{EPIC}}(R_2)(S, A, S')$, where $S, S' \sim \mathcal{D}_\mathcal{S}$ and $A \sim \mathcal{D}_\mathcal{A}$, and where the "Pearson distance" between two random variables $X$ and $Y$ be defined as $\sqrt{(1 - \rho(X,Y))/2}$, where $\rho$ denotes the Pearson correlation. Recall also that $C^{\text{EPIC}}(R)(s, a, s')$ is equal to

$$
R(s, a, s') + \mathbb{E}[\gamma R(s', A, S') - R(s, A, S') - \gamma R(S, A, S')].
$$

For the sake of brevity, let $R_1^C = C^{\text{EPIC}}(R_1)$ and $R_2^C = C^{\text{EPIC}}(R_2)$, and let $\mathcal{D}$ be the distribution over $\mathcal{S} \times \mathcal{A} \times \mathcal{S}$ given by

$$
\mathbb{P}_{(S,A,S') \sim \mathcal{D}}(S = s, A = a, S' = s) = \mathbb{P}_{S \sim \mathcal{D}_\mathcal{S}}(S = s) \cdot \mathbb{P}_{A \sim \mathcal{D}_\mathcal{A}}(A = a) \cdot \mathbb{P}_{S' \sim \mathcal{D}_\mathcal{S}}(S' = s').
$$

Moreover, let $X = R_1^C(T)$ and $Y = R_2^C(T')$, where $T$ and $T'$ are random transitions distributed according to $\mathcal{D}$. We now have that $D^{\text{EPIC}}(R_1, R_2) = \sqrt{(1 - \rho(X,Y))/2}$, where $\rho$ is the Pearson correlation.

Next, recall that the Pearson correlation between two random variables $X$ and $Y$ is defined as

$$
\frac{\mathbb{E}[(X - \mathbb{E}[X])(Y - \mathbb{E}[Y])]}{\sigma_X \sigma_Y},
$$

where $\sigma_X$ and $\sigma_Y$ are the standard deviations of $X$ and $Y$. Recall also that the standard deviation $\sigma_X$ of a random variable $X$ is equal to $\sqrt{\mathbb{E}[X^2] - \mathbb{E}[X]^2}$.

Next, note that we in this case have that $\mathbb{E}[X]$ and $\mathbb{E}[Y]$ both are equal to $0$. This follows from the way that these variables were defined, together with the linearity of expectation. Therefore, we can rewrite $\rho(X, Y)$ as

$$\frac{\mathbb{E}[XY]}{\sqrt{\mathbb{E}[X^2]}\sqrt{\mathbb{E}[Y^2]}}.$$

Let $W$ be the $(|\mathcal{S}||\mathcal{A}||\mathcal{S}|) \times (|\mathcal{S}||\mathcal{A}||\mathcal{S}|)$-dimensional diagonal matrix in which the diagonal value that corresponds to transition $t$ is equal to $\sqrt{\mathcal{P}_{T \sim \mathcal{D}}(T = t)}$. We now have that $\sqrt{\mathbb{E}[X^2]} = L_2(WR_1^C)$ and $\sqrt{\mathbb{E}[Y^2]} = L_2(WR_2^C)$. Moreover, we also have that $\mathbb{E}[XY] = (WR_1^C) \cdot (WR_2^C)$. Next, recall that the dot product $v \cdot w$ between two vectors $v$ and $w$ can be written as $L_2(v) \cdot L_2(w) \cdot \cos(\theta)$, where $\theta$ is the angle between $v$ and $w$. This means that we can rewrite $\rho(X, Y)$ as

$$\frac{L_2(WR_1^C) \cdot L_2(WR_2^C) \cdot \cos(\theta)}{L_2(WR_1^C) \cdot L_2(WR_2^C)} = \cos(\theta),$$

where $\theta$ is the angle between $WR_1^C$ and $WR_2^C$.

Since the angle between two vectors is unaffected by the scale of those vectors, we have that $\theta$ is also the angle between $WR_1^C/L_2(WR_1^C)$ and $WR_2^C/L_2(WR_2^C)$. We can now apply the Law of Cosines, and conclude that the $L_2$-distance between $WR_1^C/L_2(WR_1^C)$ and $WR_2^C/L_2(WR_2^C)$ is equal to $\sqrt{2 - 2\cos(\theta)} = \sqrt{2 - 2\rho(X, Y)} = 2 \cdot D^{\text{EPIC}}(R_1, R_2)$. This means that

$$D^{\text{EPIC}}(R_1, R_2) = \frac{1}{2} \cdot L_2\left(\frac{WR_1^C}{L_2(WR_1^C)}, \frac{WR_2^C}{L_2(WR_2^C)}\right).$$

Rewriting this completes the proof. □

The fact that EPIC can be expressed in this form is also asserted in Gleave et al. (2020), but a proof is not given. We have provided the proof here, to make the equivalence more accessible and intuitive. It is worth noting that this equivalence only holds when the "coverage distribution" is the same as the distribution used to compute $C^{\text{EPIC}}$, which is left somewhat ambiguous in Gleave et al. (2020).

## F    MAIN PROOFS

In this section, we give the proofs of all our results. We have divided it into four parts. In the first part, we prove that STARC metrics are sound. In the second part, we prove that STARC metrics are complete. In the third part, we prove that EPIC (and a class of similar metrics) all are subject to the results discussed in Appendix A. In the final part, we prove a few remaining theorems.

### F.1    SOUNDNESS

Before we can give the proof of Theorem 1, we will first state and prove several supporting lemmas.

**Lemma 1.** *For any reward functions $R_1$ and $R_2$, and any policy $\pi$, we have that*

$$|J_1(\pi) - J_2(\pi)| \leqslant \left(\frac{1}{1 - \gamma}\right) L_\infty(R_1, R_2).$$

*Proof.* This follows from straightforward algebra:

$$
\begin{aligned}
|J_1(\pi) - J_2(\pi)| = & \left| \mathbb{E}_{\xi \sim \pi} \left[ \sum_{t=0}^{\infty} \gamma^t R_1(S_t, A_t, S_{t+1}) \right] \right. \\
& \left. - \mathbb{E}_{\xi \sim \pi} \left[ \sum_{t=0}^{\infty} \gamma^t R_2(S_t, A_t, S_{t+1}) \right] \right| \\
= & \sum_{t=0}^{\infty} \gamma^t |\mathbb{E}_{\xi \sim \pi}[R_1(S_t, A_t, S_{t+1}) - R_2(S_t, A_t, S_{t+1})]| \\
\leqslant & \sum_{t=0}^{\infty} \gamma^t \mathbb{E}_{\xi \sim \pi}[|R_1(S_t, A_t, S_{t+1}) - R_2(S_t, A_t, S_{t+1})|] \\
\leqslant & \sum_{t=0}^{\infty} \gamma^t L_\infty(R_1, R_2) = \left( \frac{1}{1-\gamma} \right) L_\infty(R_1, R_2).
\end{aligned}
$$

Here the second line follows from the linearity of expectation, and the third line follows from Jensen's inequality. $\qquad \square$

Thus, the $L_\infty$-distance between two rewards bounds the difference between their policy evaluation functions. Since all norms are bilipschitz equivalent on any finite-dimensional vector space, this extends to all norms:

**Lemma 2.** *If $p$ is a norm, then there is a positive constant $K_p$ such that, for any reward functions $R_1$ and $R_2$, and any policy $\pi$, $|J_1(\pi) - J_2(\pi)| \leqslant K_p \cdot p(R_1, R_2)$.*

*Proof.* If $p$ and $q$ are norms on a finite-dimensional vector space, then there are constants $k$ and $K$ such that $k \cdot p(x) \leqslant q(x) \leqslant K \cdot p(x)$. Since $\mathcal{S}$ and $\mathcal{A}$ are finite, $\mathcal{R}$ is a finite-dimensional vector space. This means that there is a constant $K$ such that $L_\infty(R_1, R_2) \leqslant K \cdot p(R_1, R_2)$. Together with Lemma 1, this implies that

$$
|J_1(\pi) - J_2(\pi)| \leqslant \left( \frac{1}{1-\gamma} \right) \cdot K \cdot m(R_1, R_2).
$$

Letting $K_p = \left( \frac{K}{1-\gamma} \right)$ completes the proof. $\qquad \square$

Note that the constant $K_p$ given by Lemma 2 may not be the smallest value of $K$ for which this statement holds of a given norm $p$. This fact can be used to compute tighter bounds for particular STARC-metrics.

**Lemma 3.** *Let $R_1$ and $R_2$ be reward functions, and $\pi_1, \pi_2$ be two policies. If $|J_1(\pi) - J_2(\pi)| \leqslant U$ for $\pi \in \{\pi_1, \pi_2\}$, and if $J_2(\pi_2) \geqslant J_2(\pi_1)$, then*

$$
J_1(\pi_1) - J_1(\pi_2) \leqslant 2 \cdot U.
$$

*Proof.* First note that $U$ must be non-negative. Next, note that if $J_1(\pi_1) < J_1(\pi_2)$ then $J_1(\pi_1) - J_1(\pi_2) < 0$, and so the lemma holds. Now consider the case when $J_1(\pi_1) \geqslant J_1(\pi_2)$:

$$
\begin{aligned}
J_1(\pi_1) - J_1(\pi_2) = & J_1(\pi_1) - J_2(\pi_2) + J_2(\pi_2) - J_1(\pi_2) \\
\leqslant & |J_1(\pi_1) - J_2(\pi_2)| + |J_2(\pi_2) - J_1(\pi_2)|
\end{aligned}
$$

Our assumptions imply that $|J_2(\pi_2) - J_1(\pi_2)| \leqslant U$. We will next show that $|J_1(\pi_1) - J_2(\pi_2)| \leqslant U$ as well. Our assumptions imply that

$$
\begin{aligned}
& |J_1(\pi_1) - J_2(\pi_1)| \leqslant U \\
\implies & J_2(\pi_1) \geqslant J_1(\pi_1) - U \\
\implies & J_2(\pi_2) \geqslant J_1(\pi_1) - U
\end{aligned}
$$

Here the last implication uses the fact that $J_2(\pi_2) \geqslant J_2(\pi_1)$. A symmetric argument also shows that $J_1(\pi_1) \geqslant J_2(\pi_2) - U$ (recall that we assume that $J_1(\pi_1) \geqslant J_1(\pi_2)$). Together, this implies that $|J_1(\pi_1) - J_2(\pi_2)| \leqslant U$. We have thus shown that if $J_1(\pi_1) \geqslant J_1(\pi_2)$ then

$$|J_1(\pi_1) - J_2(\pi_2)| + |J_2(\pi_2) - J_1(\pi_1)| \leqslant 2 \cdot U,$$

and so the lemma holds. This completes the proof. $\qquad\square$

**Lemma 4.** *For any linear function $c : \mathbb{R}^n \to \mathbb{R}^n$ and any norm $n$, there is a positive constant $K_n$ such that $n(c(v)) \leqslant K_n \cdot n(v)$ for all $v \in \mathbb{R}^n$.*

*Proof.* First consider the case when $n(v) > 0$. In this case, we can find an upper bound for $n(c(v))$ in terms of $n(v)$ by finding an upper bound for $\frac{n(c(R))}{n(R)}$. Since $c$ is linear, and since $n$ is absolutely homogeneous, we have that for any $v \in \mathbb{R}^n$ and any non-zero $\alpha \in \mathbb{R}$,

$$\frac{n(c(\alpha \cdot v))}{n(\alpha \cdot v)} = \left(\frac{\alpha}{\alpha}\right) \frac{n(c(v))}{n(v)} = \frac{n(c(v))}{n(v)}.$$

In other words, $\frac{n(c(v))}{n(v)}$ is unaffected by scaling of $v$. We may thus restrict our attention to the unit ball of $n$. Next, since the surface of the unit ball of $n$ is a compact set, and since $\frac{n(c(v))}{n(v)}$ is continuous on this surface, the extreme value theorem implies that $\frac{n(c(v))}{n(v)}$ must take on some maximal value $K_n$ on this domain. Together, the above implies that $n(c(v)) \leqslant K_n \cdot n(v)$ for all $R$ such that $n(v) > 0$.

Next, suppose $n(v) = 0$. In this case, $v$ is the zero vector. Since $c$ is linear, this implies that $c(v) = v$, which means that $n(c(v)) = 0$ as well. Therefore, if $n(v) = 0$, then the statement holds for any $K_n$. In particular, it holds for the value $K_n$ selected above. $\qquad\square$

**Lemma 5.** *Let $c$ be a linear function $c : \mathcal{R} \to \mathcal{R}$, and let $n$ be a norm on $\mathrm{Im}(c)$. Let $R$ be any reward function, let $R_C = c(R)$, and let $R_S = \left(\frac{R_C}{n(R_C)}\right)$ if $n(R_C) > 0$, and $R_C$ otherwise. Assume there is a constant $B$ such that $J_C(\pi) = J(\pi) + B$ for all $\pi$. Then $J(\pi_1) - J(\pi_2) = n(c(R)) \cdot J_S(\pi_1) - J_S(\pi_2)$.*

*Proof.* Let us first consider the case where $n(R) = 0$. Since $n$ is a norm, $R$ must be the reward function that is 0 everywhere. Since $c$ is linear, this also implies that $n(c(R)) = 0$. In that case, both $J(\pi_1) - J(\pi_2) = 0$ for all $\pi_1$ and $\pi_2$, and $n(c(R)) \cdot x = 0$ for all $x$. Therefore, the statement holds.

Let us next consider the case when $n(R) > 0$. Since $c$ is linear, this means that $n(c(R)) > 0$. Moreover, since $R_S = R_C/n(R_C)$, and since $J_C(\pi) = J(\pi) + B$, we have that

$$J_S(\pi) = \left(\frac{1}{n(c(R))}\right)(J(\pi) + B)$$

This further implies that

$$J_S(\pi_1) - J_S(\pi_2) = \left(\frac{1}{n(c(R))}\right)(J(\pi_1) - J(\pi_2))$$

since the $B$-terms cancel out. By rearranging, we get that

$$J(\pi_1) - J(\pi_2) = n(c(R))(J_S(\pi_1) - J_S(\pi_2)).$$

This completes the proof. $\qquad\square$

**Lemma 6.** *If $R_1$ and $R_2$ differ by potential shaping with $\Phi$, then for any $\tau$ and $\mu_0$, we have that $J_2(\pi) = J_1(\pi) - \mathbb{E}_{S_0 \sim \mu_0}[\Phi(S_0)]$. Moreover, if $R_1$ and $R_2$ differ by potential shaping with $\Phi$ and $S'$-redistribution for $\tau$, then for any $\mu_0$, we have that $J_2(\pi) = J_1(\pi) - \mathbb{E}_{S_0 \sim \mu_0}[\Phi(S_0)]$.*

*Proof.* The first part follows from Lemma B.1 in Skalse et al. (2022a). The second part then follows straightforwardly from the properties of $S'$-redistribution. $\qquad\square$

**Theorem 1.** *Any STARC metric is sound.*

*Proof.* Consider any transition function $\tau$ and any initial state distribution $\mu_0$, and let $d$ be a STARC metric. We wish to show that there exists a positive constant $U$, such that for any $R_1$ and $R_2$, and any pair of policies $\pi_1$ and $\pi_2$ such that $J_2(\pi_2) \geqslant J_2(\pi_1)$, we have that

$$J_1(\pi_1) - J_1(\pi_2) \leqslant (\max_\pi J_1(\pi) - \min_\pi J_1(\pi)) \cdot K_d \cdot d(R_1, R_2).$$

Recall that $d(R_1, R_2) = m(s(R_1), s(R_2))$, where $m$ is an admissible metric. Since $m$ is admissible, we have that $p(s(R_1), s(R_2)) \leqslant K_m \cdot m(s(R_1), s(R_2))$ for some norm $p$ and constant $K_m$. Moreover, since $p$ is a norm, we can apply Lemma 2 to conclude that there is a constant $K_p$ such that for any policy $\pi$, we have that

$$|J_1^S(\pi) - J_2^S(\pi)| \leqslant K_p \cdot p(s(R_1), s(R_2)),$$

where $J_1^S$ is the policy evaluation function of $s(R_1)$, and $J_2^S$ is the policy evaluation function of $s(R_2)$. Combining this with the fact that $p(s(R_1), s(R_2)) \leqslant K_m \cdot m(s(R_1), s(R_2))$, we get

$$\begin{aligned}
|J_1^S(\pi) - J_2^S(\pi)| &\leqslant K_p \cdot p(s(R_1), s(R_2)) \\
&\leqslant K_p \cdot K_m \cdot m(s(R_1), s(R_2)) \\
&= K_{mp} \cdot d(R_1, R_2)
\end{aligned}$$

where $K_{mp} = K_p \cdot K_m$. We have thus established that, for any $\pi$, we have

$$|J_1^S(\pi) - J_2^S(\pi)| \leqslant K_{mp} \cdot d(R_1, R_2).$$

Let $\pi_1$ and $\pi_2$ be any two policies such that $J_2(\pi_2) \geqslant J_2(\pi_1)$. Note that $J_2(\pi_2) \geqslant J_2(\pi_1)$ if and only if $J_2^S(\pi_2) \geqslant J_2^S(\pi_1)$. We can therefore apply Lemma 3 and conclude that

$$J_1^S(\pi_1) - J_1^S(\pi_2) \leqslant 2 \cdot K_{mp} \cdot d(R_1, R_2).$$

By Lemma 6, there is a constant $B$ such that $J_1^C = J_1 + B$. We can therefore apply Lemma 5:

$$J_1(\pi_1) - J_1(\pi_2) \leqslant n(c(R_1)) \cdot 2 \cdot K_{mp} \cdot d(R_1, R_2).$$

We have that $n$ is a norm on $\mathrm{Im}(c)$. Moreover, $\max_\pi J_1(\pi) - \min_\pi J_1(\pi)$ is also a norm on $\mathrm{Im}(c)$ (Proposition 4). Since $\mathrm{Im}(c)$ is a finite-dimensional vector space, this means that there is a constant $K_s$ such that $n(c(R_1)) \leqslant K_s \cdot (\max_\pi J_1(\pi) - \min_\pi J_1(\pi))$ for all $R_1 \in \mathcal{R}$. Let $U = 2 \cdot K_{mp} \cdot K_s$. We have now established that, for any $\pi_1$ and $\pi_2$ such that $J_2(\pi_2) \geqslant J_2(\pi_1)$, we have

$$J_1(\pi_1) - J_1(\pi_2) \leqslant (\max_\pi J_1(\pi) - \min_\pi J_1(\pi)) \cdot U \cdot d(R_1, R_2).$$

This completes the proof. $\qquad\square$

## F.2 COMPLETENESS

In this section we give the proofs that concern completeness. We will need the following lemma:

**Lemma 7.** *Let $S \subset \mathbb{R}^n$ be the boundary of a bounded convex set whose interior includes the origin. Then there is an $\alpha > 0$ such that for any $x, y \in S$, the angle between $x$ and $y - x$ is at least $\alpha$.*

*Proof.* Let $L$ be the largest sphere which is centred around the origin, and whose interior does not intersect $S$. Note that since the interior of $S$ contains the origin, the radius of $L$ is positive. Similarly, let $U$ be the smallest sphere which is centred around the origin, and whose exterior does not intersect $S$. (In other words, $L$ and $U$ are two spheres such that $S$ lies "between" $L$ and $U$. Note that if $S$ is a sphere centred around the origin, then $L = U$.)

Let $x$ and $y$ be two arbitrary points in $S$, and let $\theta$ be the angle between $-x$ and $y - x$:

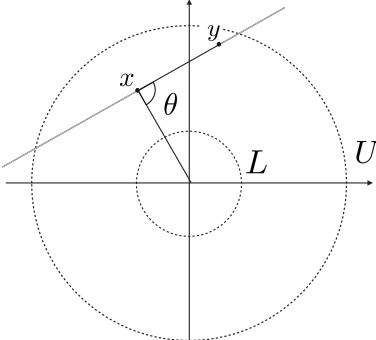

Consider the line that passes through $x$ and $y$. This line cannot intersect the interior of $L$, since $S$ is the boundary of a *convex* set. Note also that $\theta$ gets bigger if we reduce the magnitude of $x$. Thus, let $x'$ be the vector that results from reducing the magnitude of $x$ until the line between $x'$ and $y$ is a tangent of $L$:

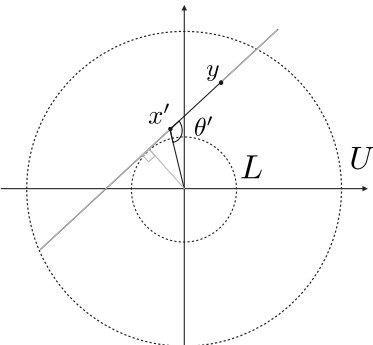

Now the angle $\theta$ between $-x$ and $y - x$ is at most as big as the angle $\theta'$ between $-x'$ and $y - x'$. Next, let the point where the line between $x'$ and $y$ intersects $L$ be called $A$, and the point where it intersects $U$ be called $B$. Consider the line segment between $A$ and $B$:

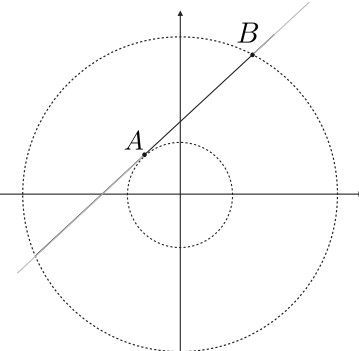

This line segment is a compact set, which means that there is a point $x''$ along this line which maximises the angle between $-x''$ and $y - x''$ (note that this point in fact is equal to $B$, but we will not need this fact in our proof). Let this angle be $\theta''$. We now have that $\theta > \theta' > \theta''$. Moreover, the value of $\theta''$ does not depend on $x$ or $y$, which means that the angle $\theta$ between $-x$ and $y - x$ is at most $\theta$ for all points $x, y \in S$. This in turn means that the angle between $x$ and $y - x$ is at least $\alpha = \pi - \theta$, which completes the proof. $\qquad\square$

Using this, we can now show that we can get a lower bound on the *angle* between two standardised reward functions in terms of their STARC-distance:

**Lemma 8.** *For any STARC metric d, there exist an $\ell_1 \in \mathbb{R}^+$ such that the angle $\theta$ between $s(R_1)$ and $s(R_2)$ satisfies $\ell_1 \cdot d(R_1, R_2) \leqslant \theta$ for all $R_1$, $R_2$ for which neither $s(R_1)$ or $s(R_2)$ is 0.*

*Proof.* Let $d$ be an arbitrary STARC-metric, and let $R_1$ and $R_2$ be two arbitrary reward functions for which neither $s(R_1)$ or $s(R_2)$ is 0. Recall that $d(R_1, R_2) = m(s(R_1), s(R_2))$, where $m$ is a metric that is bilipschitz equivalent to some norm. Since all norms are bilipschitz equivalent on any finite-dimensional vector space, this means that $m$ is bilipschitz equivalent to the $L_2$-norm. Thus, there are positive constants $p$, $q$ such that

$$p \cdot m(s(R_1), s(R_2)) \leqslant L_2(s(R_1), s(R_2)) \leqslant q \cdot m(s(R_1), s(R_2)).$$

In particular, the $L_2$-distance between $s(R_1)$ and $s(R_2)$ is at least $\epsilon = p \cdot d(R_1, R_2)$. For the rest of our proof, it will be convenient to assume that $\epsilon < L_2(s(R_1))$; this can be ensured by picking a $p$ that is sufficiently small.

Let us plot the plane which contains $s(R_1)$, $s(R_2)$, and the origin, and orient it so that $s(R_1)$ points straight up, and so that $s(R_2)$ is not on the left-hand side:

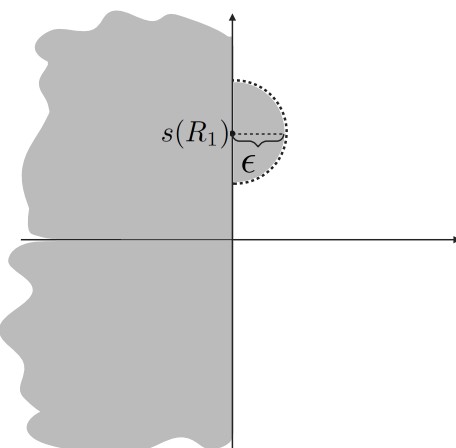

Since the distance between $s(R_1)$ and $s(R_2)$ is at least $\epsilon$, and since $s(R_2)$ is not on the left-hand side, we know that $s(R_2)$ cannot be inside of the region shaded grey in the figure above (though it may be on the boundary). Moreover, as per Lemma 7, we know that the angle between $s(R_1)$ and $s(R_2) - s(R_1)$ is at least $\alpha$, where $\alpha > 0$. This means that we also can rule out the following region:

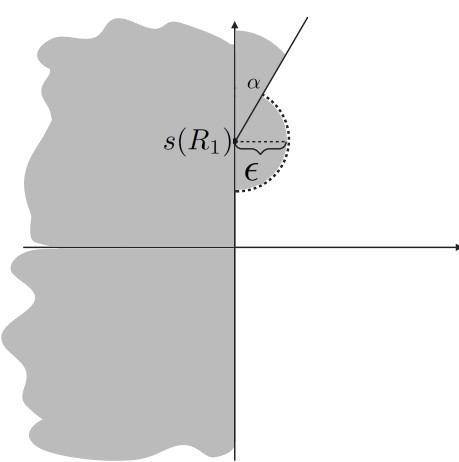

Moreover, let $v$ be the element of $\mathrm{Im}(s)$ that is perpendicular to $s(R_1)$, lies on a plane with $s(R_1)$, $s(R_2)$, and the origin, and points in the same direction as $s(R_2)$ within this plane. Since $\mathrm{Im}(s)$ is convex, we know that $s(R_2)$ cannot lie within the triangle formed by the $x$-axis, the $y$-axis, and the line between $s(R_1)$ and $v$:

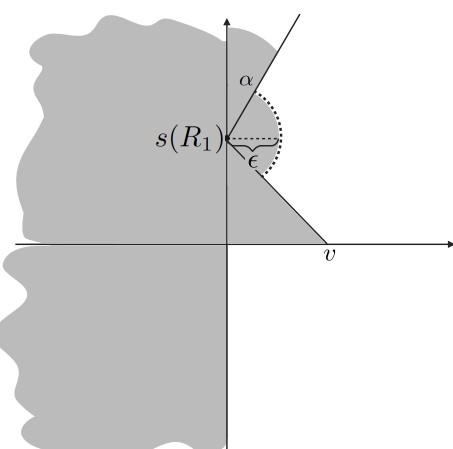

Since $\mathrm{Im}(s)$ is closed and convex, we know that there is a vector $a$ in $\mathrm{Im}(s)$ whose $L_2$-norm is bigger than all other vectors in $\mathrm{Im}(s)$, and a (non-zero) vector $b$ in $\mathrm{Im}(s)$ whose $L_2$-norm is smaller than all other (non-zero) vectors in $\mathrm{Im}(s)$. From this, we can infer that the angle between $s(R_1)$ and $v - s(R_1)$ is at least $\beta = \arctan(b/a)$. Also note that $\beta > 0$.

We now have everything we need to derive a lower bound on the angle $\theta$ between $s(R_1)$ and $s(R_2)$. First note that this angle can be no greater than the angle between $s(R_1)$ and the points marked $A$ and $B$ in the figure below (whichever is smaller):

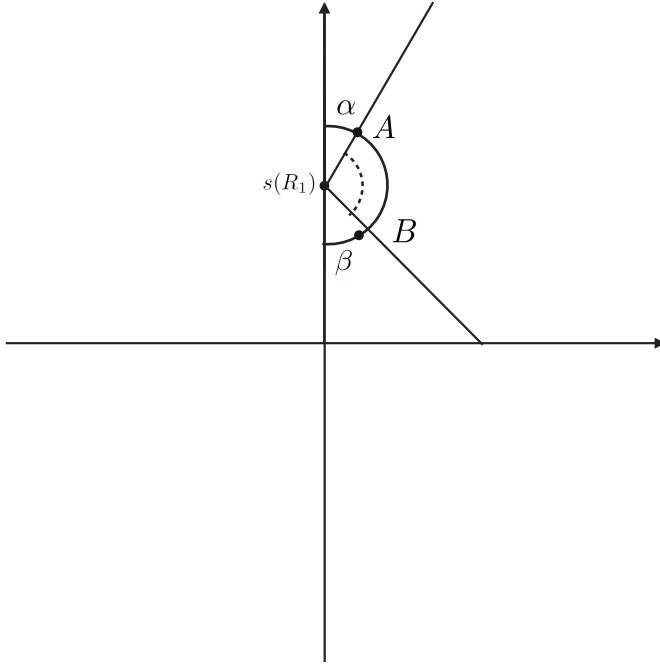

To make things easier, replace both $\alpha$ and $\beta$ with $\gamma = \min(\alpha, \beta)$. Since this makes the shaded region smaller, we still have that $s(R_2)$ cannot be in the interior of the new shaded region. Moreover, in this case, we know that the angle between $s(R_1)$ and $s(R_2)$ is no smaller than the angle $\theta'$ between $s(R_1)$ and the point marked $A$:

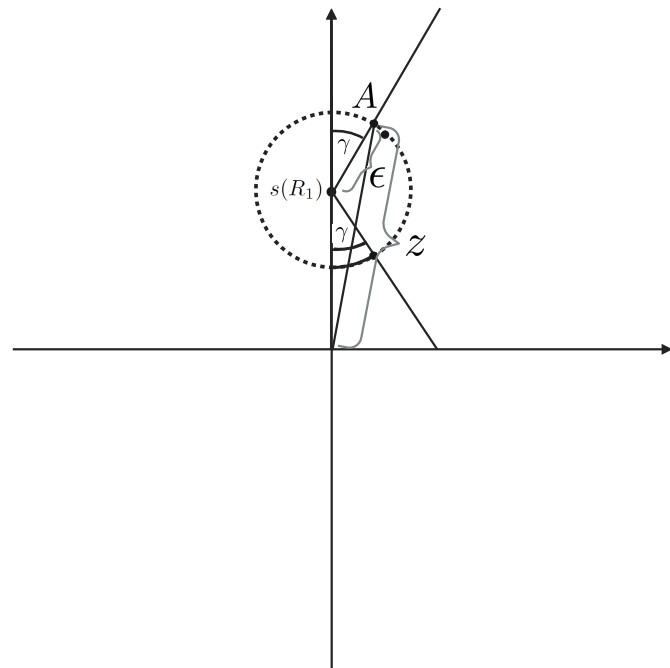

Deriving this angle is now just a matter of trigonometry. Letting $z$ denote $L_2(A)$, we have that:

$$\frac{\epsilon}{\sin(x)} = \frac{z}{\sin(\pi - \gamma)} = \frac{z}{\sin(\gamma)}$$

From this, we get that

$$\theta' = \arcsin\left(\left(\frac{\epsilon}{z}\right)\sin(\gamma)\right)$$
$$\geq \left(\frac{\epsilon}{z}\right)\sin(\gamma)$$

Moreover, it is also straightforward to find an upper bound $z'$ for $z$. Specifically, we have that $z^2 = L_2(s(R_1))^2 + \epsilon^2 - 2L_2(s(R_1))\epsilon\cos(\pi - \gamma)$. Since $\epsilon < L_2(s(R_1))$, this means that

$$z < \sqrt{2L_2(s(R_1))^2 - 2L_2(s(R_1))^2\cos(\pi - \gamma)}.$$

Moreover, since $\mathrm{Im}(s)$ is compact, there is a vector $a$ in $\mathrm{Im}(s)$ whose $L_2$-norm is bigger than all other vectors in $\mathrm{Im}(s)$. We thus know that

$$z < z' = \sqrt{2L_2(a)^2 - 2L_2(a)^2\cos(\pi - \gamma)}.$$

Putting this together, we have that

$$\theta \geq \theta' \geq \left(\frac{\epsilon}{z'}\right)\sin(\gamma) = m(s(R_1), s(R_2)) \cdot p \cdot \left(\frac{\sin(\gamma)}{z'}\right).$$

Setting $\ell_1 = p \cdot \left(\frac{\sin(\gamma)}{z'}\right)$ thus completes the proof. $\qquad\square$

Finally, before we can give the full proof, we will also need the following:

**Lemma 9.** *For any invertible matrix $M : \mathbb{R}^n \to \mathbb{R}^n$ there is an $\ell_2 \in (0, 1]$ such that for any $v, w \in \mathbb{R}^n$, the angle $\theta'$ between $Mv$ and $Mw$ satisfies $\theta' \geqslant \ell_2 \cdot \theta$, where $\theta$ is the angle between $v$ and $w$.*

*Proof.* We will first prove that this holds in the 2-dimensional case, and then extend this proof to the general $n$-dimensional case.

Let $M$ be an arbitrary invertible matrix $\mathbb{R}^2 \to \mathbb{R}^2$. First note that we can factor $M$ via Singular Value Decomposition into three matrices $U, \Sigma, V$, such that $M = U\Sigma V^\top$, where $U$ and $V$ are orthogonal matrices, and $\Sigma$ is a diagonal matrix with non-negative real numbers on the diagonal. Since $M$ is invertible, we also have that $\Sigma$ cannot have any zeroes along its diagonal. Next, recall that orthogonal matrices preserve angles. This means that we can restrict our focus to just $\Sigma$.[6]

Let $\alpha$ and $\beta$ be the singular values of $M$. We may assume, without loss of generality, that

$$\Sigma = \begin{pmatrix} \alpha & 0 \\ 0 & \beta \end{pmatrix}.$$

Moreover, since scaling the $x$ and $y$-axes uniformly will not affect the angle between any vectors after multiplication, we can instead equivalently consider the matrix

$$\Sigma = \begin{pmatrix} \alpha/\beta & 0 \\ 0 & 1 \end{pmatrix}.$$

Let $v, w \in \mathbb{R}^2$ be two arbitrary vectors with angle $\theta$, and let $\theta'$ be the angle between $\Sigma v$ and $\Sigma w$. We will derive a lower bound on $\theta'$ expressed in terms of $\theta$. Moreover, since the angle between $v$ and $w$ is not affected by their magnitude, we will assume (without loss of generality) that both $v$ and $w$ have length 1 (under the $L_2$-norm).

First, note that if $\theta = \pi$ then $v = -w$. This means that $\Sigma v = -\Sigma w$, since $\Sigma$ is a linear transformation, which in turn means that $\theta' = \pi$. Thus $\theta' \geqslant \ell_2 \cdot \theta$ as long as $\ell_2 \leqslant 1$. Next, assume that $\theta < \pi$.

We may assume (without loss of generality) that the angle between $v$ and the $x$-axis is no bigger than the angle between $w$ and the $x$-axis. Let $\phi$ be the angle between the $x$-axis and the vector that is in the middle between $v$ and $w$. This means that we can express $v$ as $(\cos(\phi - \theta/2), \sin(\phi - \theta/2))$ and $w$ as $(\cos(\phi + \theta/2), \sin(\phi + \theta/2))$. Moreover, since reflection along either of the axes will not change the angle between either $v$ and $w$ or $\Sigma v$ and $\Sigma w$, we may assume (without loss of generality) that $\phi \in [0, \pi/2]$. For convenience, let $\sigma = \alpha/\beta$.

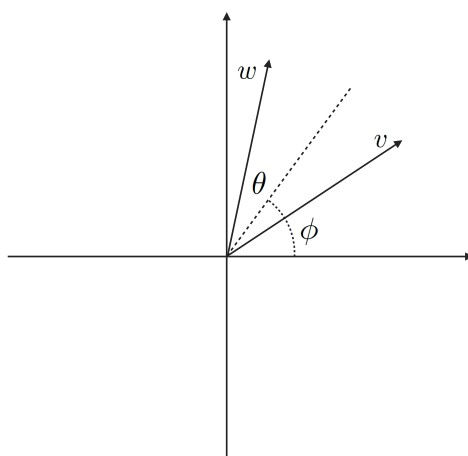

---

[6]If there are vectors $x, y$ such that the angle between $x$ and $y$ is $\theta$ and the angle between $Mx$ and $My$ is $\theta'$, then there are vectors $v, w$ such that the angle between $x$ and $y$ is $\theta$ and the angle between $\Sigma v$ and $\Sigma w$ is $\theta'$, and vice versa.

(Note that we can visualise the action of $\Sigma$ as scaling the $x$-axis in the figure above by $\sigma$.)

We now have that $\Sigma v = (\sigma \cos(\phi - \theta/2), \sin(\phi - \theta/2))$ and $\Sigma w = (\sigma \cos(\phi + \theta/2), \sin(\phi + \theta/2))$. Using the dot product, we get that

$$\cos(\theta') = \frac{\sigma^2 \cos(\phi - \theta/2) \cos(\phi + \theta/2) + \sin(\phi - \theta/2) \sin(\phi + \theta/2)}{\sqrt{\sigma^2 \cos^2(\phi - \theta/2) + \sin^2(\phi - \theta/2)}\sqrt{\sigma^2 \cos^2(\phi + \theta/2) + \sin^2(\phi + \theta/2)}}.$$

We next note that if $\theta \in [0, \pi)$ and $\phi \in [0, \pi/2]$, then the derivative of this expression with respect to $\phi$ can only be 0 when $\phi \in \{0, \pi/2\}$.[7] This means that $\cos(\theta')$ must be maximised or minimised when $\phi$ is either 0 or $\pi/2$, which in turn means that the angle $\theta'$ must be minimised or maximised when $\phi$ is either 0 or $\pi/2$.

It is now easy to see that if $\sigma > 1$ then $\theta'$ is minimised when $\phi = 0$, and that if $\sigma < 1$ then $\theta'$ is minimised when $\phi = \pi/2$. Moreover, if $\phi = \pi/2$, then

$$\theta' = 2\arctan\left(\frac{\sigma \cos(\pi/2 - \theta/2)}{\sin(\pi/2 - \theta/2)}\right) = 2\arctan\left(\sigma \tan(\theta/2)\right),$$

which in turn is greater than $\theta \cdot \sigma$ when $\sigma < 1$. Similarly, if $\phi = 0$, then

$$\theta' = 2\arctan\left(\frac{\sin(\theta/2)}{\sigma \cos(\theta/2)}\right) = 2\arctan\left(\sigma^{-1} \tan(\theta/2)\right),$$

which is in turn greater than $\sigma^{-1} \cdot \theta$ when $\sigma > 1$. In either case, we thus have that

$$\theta' \geqslant \theta \cdot \min(\sigma, \sigma^{-1}) = \theta \cdot \min(\beta/\alpha, \alpha/\beta).$$

We have therefore show that, for any invertible matrix $M : \mathbb{R}^2 \to \mathbb{R}^2$, there exists a positive constant $\min(\beta/\alpha, \alpha/\beta)$, where $\alpha$ and $\beta$ are the singular values of $M$, such that if $v, w \in \mathbb{R}^2$ have angle $\theta$, then the angle between $Mv$ and $Mw$ is at least $\theta \cdot \min(\beta/\alpha, \alpha/\beta)$.

To generalise this to the general $n$-dimensional case, let $v, w \in \mathbb{R}^n$ be two arbitrary vectors. Consider the 2-dimensional linear subspace given by $S = \mathrm{span}(v, w)$, and note that $M(S)$ also is a 2-dimensional linear subspace of $\mathbb{R}^n$ (since $M$ is linear and invertible). The linear transformation which $M$ induces between $S$ and $M(S)$ is isomorphic to a linear transformation $M' : \mathbb{R}^2 \to \mathbb{R}^2$.[8] We can thus apply our previous result for the two-dimensional case, and conclude that if the angle between $v$ and $w$ is $\theta$, then the angle between $Mv$ and $Mw$ is at least $\theta \cdot \min(\beta/\alpha, \alpha/\beta)$, where $\alpha$ and $\beta$ are the singular values of $M'$. Next, note that the singular values of $M'$ cannot be smaller than the smallest singular values of $M$ or bigger than the biggest singular values of $M$. We can therefore let $\ell_2 = \alpha/\beta$, where $\alpha$ is the smallest singular value of $M$ and $\beta$ is the greatest singular value of $M$, and conclude that the angle between $Mv$ and $Mw$ must be at least $\ell_2 \cdot \theta$. Since the value of $\ell_2$ does not depend on $v$ or $w$, this completes the proof. $\qquad\square$

With these lemmas, we can now finally prove that all STARC metrics are complete:

**Theorem 2.** *Any STARC metric is complete.*

*Proof.* Let $d$ be an arbitrary STARC metric. We need to show that there exists a positive constant $L$ such that, for any reward functions $R_1$ and $R_2$, there are two policies $\pi_1, \pi_2$ with $J_2(\pi_2) \geqslant J_2(\pi_1)$ and

$$J_1(\pi_1) - J_1(\pi_2) \geqslant L \cdot \left(\max_{\pi} J_1(\pi) - \min_{\pi} J_1(\pi)\right) \cdot d(R_1, R_2),$$

and moreover, if both $\max_{\pi} J_1(\pi) - \min_{\pi} J_1(\pi) = 0$ and $\max_{\pi} J_2(\pi) - \min_{\pi} J_2(\pi) = 0$, then we have that $d(R_1, R_2) = 0$.

---

[7]For example, this may be verified using tools such as Wolfram Alpha.

[8]To see this, let $A$ be an orthonormal matrix that rotates $\mathbb{R}^2$ to align with $S$, and let $B$ be an orthonormal matrix that rotates $M(S)$ to align with $\mathbb{R}^2$. Now $M' = BMA$ is an invertible linear transformation $\mathbb{R}^2 \to \mathbb{R}^2$. Moreover, since orthonormal matrices preserve the angles between vectors, we have that $v, w \in S$ have angle $\theta$ and $Mv, Mw \in M(S)$ have angle $\theta'$, if and only if $A^{-1}v, A^{-1}w \in \mathbb{R}^2$ have angle $\theta$ and $BMv, BMw \in \mathbb{R}^2$ have angle $\theta'$. Note that $M'A^{-1}v = BMv$ and $M'A^{-1}w = BMw$. This means that there are $v, w \in S$ such that $v, w$ have angle $\theta$ and $Mv, Mw$ have angle $\theta'$, if and only if there are $v', w' \in \mathbb{R}^2$ such that $v', w'$ have angle $\theta$ and $M'v'$ and $M'w'$ have angle $\theta'$ (with $v' = A^{-1}v$ and $w' = A^{-1}w$).

We first note that the last condition holds straightforwardly. If $\max_\pi J_1(\pi) - \min_\pi J_1(\pi) = 0$ and $\max_\pi J_2(\pi) - \min_\pi J_2(\pi) = 0$ then $R_1$ and $R_2$ have the same policy order, which means that Proposition 6 implies that $d(R_1, R_2) = 0$. This condition is therefore satisfied.

For the first condition, first note that if $\max_\pi J_1(\pi) - \min_\pi J_1(\pi) = 0$, then the statement holds trivially for any non-negative $L$ (since the other two terms on the right-hand side of the inequality are strictly non-negative).

Let us next consider the case where both $\max_\pi J_1(\pi) - \min_\pi J_1(\pi) > 0$ and $\max_\pi J_1(\pi) - \min_\pi J_1(\pi) > 0$. We need to introduce a new definition. Let $m : \Pi \to \mathbb{R}^{|\mathcal{S}||\mathcal{A}||\mathcal{S}|}$ be the function that takes a policy $\pi$, and returns the vector where $m(\pi)[s, a, s'] = \sum_{t=0}^\infty \gamma^t \mathbb{P}(S_t, A_t, S_{t+1} = s, a, s')$, where the probability is for a trajectory sampled from $\pi$ under $\tau$ and $\mu_0$. In other words, $m$ returns the long-run discounted cumulative probability with which $\pi$ visits each transition. Next, note that $J(\pi) = m(\pi) \cdot R$. This means that $m$ can be used to decompose $J$ into two steps, the first of which is independent of the reward function, and the second of which is a linear function.

We will use $d$ to derive a lower bound on the angle $\theta$ between the level sets of $J_1$ and $J_2$ in $\mathrm{Im}(m)$. We will then show that $\mathrm{Im}(m)$ contains an open set with a certain diameter. From this, we can find two policies that incur a certain amount of regret.

First, by Lemma 8, there exists an $\ell_1$ such that for any non-trivial $R_1$ and $R_2$, the angle between $s(R_1)$ and $s(R_2)$ is at least $\ell_1 \cdot d(R_1, R_2)$. To make our proof easier, we will assume that we pick an $\ell_1$ that is small enough to ensure that $\ell_1 \cdot d(R_1, R_2) \leqslant \pi/2$ for all $R_1, R_2$.

Note that $s(R_1)$ and $s(R_2)$ may not be parallel with $\mathrm{Im}(m)$, which means that the angle between $s(R_1)$ and $s(R_2)$ may not be the same as the angle between the level sets of $J_1$ and $J_2$ in $\mathrm{Im}(m)$. Therefore, consider the matrix $M$ that projects $\mathrm{Im}(c)$ onto the linear subspace of $\mathcal{R}$ that is parallel to $\mathrm{Im}(m)$, where $c$ is the canonicalisation function of $d$. Now the angle between $Ms(R_1)$ and $Ms(R_2)$ is the same as the angle between the level sets of the linear functions which $J_1$ and $J_2$ induce on $\mathrm{Im}(m)$. Moreover, note that $M$ is invertible, since any two reward functions in $\mathrm{Im}(c)$ induce different policy orderings except when they differ by positive linear scaling (Proposition 1). We can therefore apply Lemma 8, and conclude that there exists an $\ell_2 \in (0, 1]$, such that the angle $\theta$ between the level sets of $J_1$ and $J_2$ in $\mathrm{Im}(m)$ is at least $\ell_2 \cdot \ell_1 \cdot d(R_1, R_2)$. Moreover, since $\ell_1 \cdot d(R_1, R_2)$ is at most $\pi/2$, and since $\ell_2 \leqslant 1$, we have that $\ell_2 \cdot \ell_1 \cdot d(R_1, R_2)$ is at most $\pi/2$.

This gives us that, for any two policies $\pi_1, \pi_2$, we have:

$$
\begin{aligned}
J_1(\pi_1) - J_1(\pi_2) &= J_1^C(\pi_1) - J_1^C(\pi_2) \\
&= c(R_1)m(\pi_1) - c(R_1)m(\pi_2) \\
&= c(R_1)(m(\pi_1) - m(\pi_2)) \\
&= M(c(R_1))(m(\pi_1) - m(\pi_2)) \\
&= L_2(M(c(R_1))) \cdot L_2(m(\pi_1) - m(\pi_2)) \cdot \cos(\phi)
\end{aligned}
$$

where $\phi$ is the angle between $M(c(R_1))$ and $m(\pi_1) - m(\pi_2)$, and $J_1^C$ is the evaluation function of $c(R_1)$. Note that the first line follows from Lemma 6. We can thus derive a lower bound on worst-case regret by deriving a lower bound for the greatest value of this expression.

We have that $\mathrm{Im}(m)$ contains a set that is open in the smallest affine space which contains $\mathrm{Im}(m)$ (see Skalse et al., 2022b). This means that there is an $\epsilon$ such that $\mathrm{Im}(m)$ contains a sphere of diameter $\epsilon$. We will show that we always can find two policies within this sphere that incur a certain amount of regret. Consider the 2-dimensional cut which goes through the middle of this sphere and is parallel with the normal vectors of the level sets of $J_1$ and $J_2$. The intersection between this cut and the $\epsilon$-sphere forms a 2-dimensional circle with diameter $\epsilon$. Let $\pi_1, \pi_2$ be the two policies for which $m(\pi_1)$ and $m(\pi_2)$ lie opposite to each other on this circle, and satisfy that $J_2(\pi_1) = J_2(\pi_2)$ (or, equivalently, that $Mc(R_1) \cdot m(\pi_1) = Mc(R_1) \cdot m(\pi_2)$). Without loss of generality, we may assume that $J_1(\pi_1) \geqslant J_1(\pi_2)$.

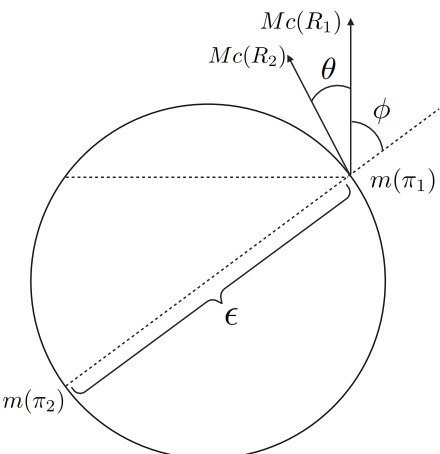

Now note that $L_2(m(\pi_1) - m(\pi_2)) = \epsilon$. Moreover, recall that the angle $\theta$ between $Mc(R_1)$ and $Mc(R_2)$ is at least $\theta' = \ell_1 \cdot \ell_2 \cdot d(R_1, R_2)$, and that this quantity is at most $\pi/2$. This means that the angle $\phi$ is at most $\pi/2 - \theta'$, and so $\cos(\phi)$ is at least $\cos(\pi/2 - \theta') = \cos(\pi/2 - \ell_1 \cdot \ell_2 \cdot d(R_1, R_2))$. This means that we have two policies $\pi_1, \pi_2$ where $J_2(\pi_2) = J_2(\pi_1)$ and such that

$$J_1(\pi_1) - J_1(\pi_2) = L_2(M(c(R_1))) \cdot L_2(m(\pi_1) - m(\pi_2)) \cos(\phi)$$
$$= L_2(M(c(R_1))) \cdot \epsilon \cdot \cos(\pi/2 - \ell_2 \cdot \ell_1 \cdot d(R_1, R_2)).$$

Note that $\cos(\pi/2 - x) \geqslant x \cdot 2/\pi$ when $x \leqslant \pi/2$, and that $\ell_2 \cdot \ell_1 \cdot d(R_1, R_2) \leqslant \pi/2$. Putting this together, we have that there must exist two policies $\pi_1, \pi_2$ with $J_2(\pi_2) = J_2(\pi_1)$ such that

$$J_1(\pi_1) - J_1(\pi_2) \geqslant L_2(M(c(R_1))) \cdot \left( \frac{\epsilon \cdot \ell_1 \cdot \ell_2 \cdot 2}{\pi} \right) \cdot d(R_1, R_2).$$

Next, note that, if $p$ is a norm and $M$ is an invertible matrix, then $p \circ M$ is also a norm. Furthermore, recall that $\max_\pi J_1(\pi) - \min_\pi J_1(\pi)$ is a norm on $\mathrm{Im}(c)$, when $c$ is a canonicalisation function (Proposition 4). Since all norms are equivalent on a finite-dimensional vector space, this means that there must exist a positive constant $\ell_3$ such that $L_2(M(c(R_1))) \geqslant \ell_3 \cdot (\max_\pi J_1(\pi) - \min_\pi J_1(\pi))$. We can therefore set $L = (\epsilon \cdot \ell_1 \cdot \ell_2 \cdot \ell_3 \cdot 2/\pi)$, and obtain the result that we want:

$$J_1(\pi_1) - J_1(\pi_2) \geqslant L \cdot (\max_\pi J_1(\pi) - \min_\pi J_1(\pi)) \cdot d(R_1, R_2).$$

Finally, we must consider the case where $R_2$ is trivial under $\tau$ and $\mu_0$, but where $R_1$ is not. In this case, $J_2(\pi_2) \geqslant J_2(\pi_1)$ for all $\pi_1$ and $\pi_2$, which means that $\max_{\pi_1, \pi_2 : J_2(\pi_2) \geqslant J_2(\pi_1)} J_1(\pi_1) - J_1(\pi_2) = \max_\pi J_1(\pi) - \min_\pi J_1(\pi)$. Therefore, the statement holds for any $L$ as long as we ensure that $L \cdot d(R_1, R_0) \leqslant 1$ for all $R_1$ and $R_2$. This completes the proof. $\qquad \square$

### F.3    Issues with EPIC, and Similar Metrics

In this appendix, we prove the results from in Appendix A. Moreover, we state and prove versions of these theorems that are more general than the versions given in the main text. First, we need a few new definitions:

**Definition 7.** A function $c : \mathcal{R} \to \mathcal{R}$ is an *EPIC-like canonicalisation function* if $c$ is linear, $c(R)$ and $R$ differ by potential shaping, and $c(R_1) = c(R_2)$ if and only if $R_1$ and $R_2$ only differ by potential shaping.

**Definition 8.** A function $d : \mathcal{R} \times \mathcal{R} \to \mathbb{R}$ is an *EPIC-like metric* if there is an EPIC-like canonicalisation function $c$, a function $n$ that is a norm on $\mathrm{Im}(c)$, and a metric $m$ that is admissible on $\mathrm{Im}(c)$, such that $d(R_1, R_2) = m(s(R_1), s(R_2))$, where $s(R) = c(R)/n(c(R))$ when $n(c(R)) \neq 0$, and $c(R)$ otherwise.

Note that $C^{\mathrm{EPIC}}$ is an EPIC-like canonicalisation function, and that EPIC is an EPIC-like metric.

**Theorem 3.** *For any EPIC-like metric $d$ there exists a positive constant $U$, such that for any reward functions $R_1$ and $R_2$, if two policies $\pi_1$ and $\pi_2$ satisfy that $J_2(\pi_2) \geqslant J_2(\pi_1)$, then we have that*

$$J_1(\pi_1) - J_1(\pi_2) \leqslant U \cdot L_2(R_1) \cdot d(R_1, R_2).$$

*Proof.* We wish to show that there is a positive constant $U$, such that for any $R_1$ and $R_2$, and any pair of policies $\pi_1$ and $\pi_2$ such that $J_2(\pi_2) \geqslant J_2(\pi_1)$, we have

$$J_1(\pi_1) - J_1(\pi_2) \leqslant U \cdot L_2(R_1) \cdot d(R_1, R_2).$$

Moreover, this must hold for any choice of $\tau$ and $\mu_0$.

Recall that $d(R_1, R_2) = m(s(R_1), s(R_2))$, where $m$ is an admissible metric. Since $m$ is admissible, we have that $p(s(R_1), s(R_2)) \leqslant K_m \cdot m(s(R_1), s(R_2))$ for some norm $p$ and constant $K_m$. Moreover, since $p$ is a norm, we can apply Lemma 2 to conclude that there is a constant $K_p$ such that for any policy $\pi$, any transition function $\tau$, and any initial state distribution $\mu_0$, we have that

$$|J_1^S(\pi) - J_2^S(\pi)| \leqslant K_p \cdot p(s(R_1), s(R_2)).$$

Combining this with the fact that $p(s(R_1), s(R_2)) \leqslant K_m \cdot m(s(R_1), s(R_2))$, we get

$$K_p \cdot p(s(R_1), s(R_2)) \leqslant K_p \cdot K_m \cdot m(s(R_1), s(R_2))$$
$$= K_{mp} \cdot d(R_1, R_2)$$

where $K_{mp} = K_p \cdot K_m$. We have thus established that, for any $\pi$, $\tau$, and $\mu_0$, we have

$$|J_1^S(\pi) - J_2^S(\pi)| \leqslant K_{mp} \cdot d(R_1, R_2).$$

Consider an arbitrary transition function $\tau$ and initial state distribution $\mu_0$, and let $\pi_1$ and $\pi_2$ be any two policies such that $J_2(\pi_2) \geqslant J_2(\pi_1)$ under $\tau$ and $\mu_0$. Note that $J_2(\pi_2) \geqslant J_2(\pi_1)$ if and only if $J_2^S(\pi_2) \geqslant J_2^S(\pi_1)$. We can therefore apply Lemma 3 and conclude that

$$J_1^S(\pi_1) - J_1^S(\pi_2) \leqslant 2 \cdot K_{mp} \cdot d(R_1, R_2).$$

By Lemma 6, there is a constant $B$ such that $J_1^C = J_1 + B$. We can therefore apply Lemma 5:

$$J_1(\pi_1) - J_1(\pi_2) \leqslant n(c(R_1)) \cdot 2 \cdot K_{mp} \cdot d(R_1, R_2).$$

By Lemma 4, there is a positive constant $K_n$ such that $n(c(R)) \leqslant K_n \cdot n(R)$ for all $R \in \mathcal{R}$.

$$J_1(\pi_1) - J_1(\pi_2) \leqslant K_n \cdot n(R_1) \cdot 2 \cdot K_{mp} \cdot d(R_1, R_2).$$

Moreover, since $n$ is a norm, and since $\mathcal{R}$ is a finite-dimensional vector space, we have that there is a constant $K_2$ such that $n(R) \leqslant K_2 \cdot L_2(R)$ for all $R \in \mathcal{R}$. Let $U = 2 \cdot K_n \cdot K_{mp} \cdot K_2$. We have now established that, for any $\pi_1$ and $\pi_2$ such that $J_2(\pi_2) \geqslant J_2(\pi_1)$, we have that

$$J_1(\pi_1) - J_1(\pi_2) \leqslant U \cdot L_2(R_1) \cdot d(R_1, R_2).$$

Note that $U$ does not depend on $\tau$ or $\mu_0$. This completes the proof. $\qquad\square$

**Theorem 4.** *No EPIC-like metric is sound.*

*Proof.* Consider an arbitrary transition function $\tau$ and an arbitrary initial state distribution $\mu_0$, and let $d$ be an EPIC-like metric with canonicalisation function $c$, normalisation function $n$, and admissible metric $m$.

Let $\mathcal{X}$ be a linear subspace of $\mathrm{Im}(c)$, such that there, for any reward function $R \in \mathrm{Im}(c)$, is exactly one reward function $R' \in \mathcal{X}$ such that $R$ and $R'$ differ by $S'$-redistribution under $\tau$.

Let $R_1$ be an arbitrary reward function in $\mathcal{X}$ such that $n(R_1) = 1$, and let $R_2 = -R_1$. Note that $R_1$ and the reward function that is $0$ everywhere do not differ by potential shaping and $S'$-redistribution – this is ensured by the fact that they are distinct, and both included in $\mathcal{X}$. As per Proposition 1, this implies that $R_1$ does not have the same policy order as the reward function that is $0$ everywhere. This, in turn, means that $\max_\pi J_1(\pi) - \min_\pi J_1(\pi) > 0$. Moreover, since $R_2 = -R_1$, this implies that, if $\pi_1$ is a policy that is optimal under $R_1$, and $\pi_2$ is a policy that is optimal under $R_2$, then $\pi_2$

is maximally bad under $R_1$, and $\pi_1$ is maximally bad under $R_2$. In other words, there are policies $\pi_1, \pi_2$ such that $J_2(\pi_2) \geqslant J_2(\pi_1)$, and

$$\frac{J_1(\pi_1) - J_1(\pi_2)}{\max_\pi J_1(\pi) - \min_\pi J_1(\pi)} = 1.$$

This is the greatest value for this expression, and so the regret for $R_1$ and $R_2$ is maximally high.

Next, let $R_1' = \epsilon \cdot R_1$, and $R_2' = \epsilon \cdot R_2$, for some small positive value $\epsilon$. Since positive linear scaling does not affect the regret, we have that the regret for $R_1'$ and $R_2'$ also is 1.

Let $x$ be a vector in $\mathrm{Im}(c)$ that is orthogonal to $\mathcal{X}$. Note that movement along $x$ corresponds to $S'$-redistribution under $\tau$. Next, let $R_1'' = R_1' + \alpha \cdot x$ and $R_2'' = R_2' + \beta \cdot x$, where $\alpha$ and $\beta$ are two positive constants such that $n(R_1'') = 1$ and $n(R_2'') = 1$. Since movement along $x$ corresponds to $S'$-redistribution under $\tau$, and since $S'$-redistribution under $\tau$ does not affect regret, we have that the regret for $R_1''$ and $R_2''$ is 1.

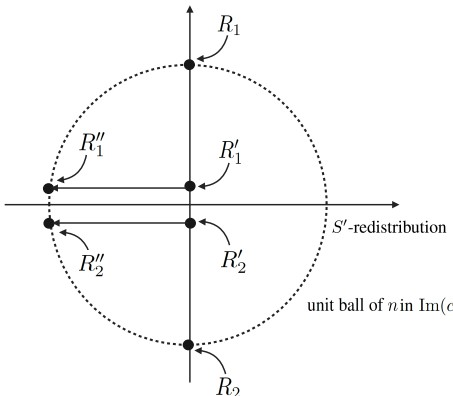

Now, since $R_1''$ and $R_2''$ are in $\mathrm{Im}(c)$, and since $n(R_1'') = 1$ and $n(R_2'') = 1$, we have that $d(R_1'', R_2'') = m(R_1'', R_2'')$. By making $\epsilon$ sufficiently small, we can ensure that this value is arbitrarily close to 0. Therefore, for any simple STARC metric $d$ and any environment, there are reward functions such that $R_1''$ and $R_2''$ have maximally high regret, but $d(R_1'', R_2'')$ is arbitrarily close to 0. $\square$

**Theorem 5.** *There exist reward functions $R_1$, $R_2$ such that $d(R_1, R_2) > 0$ for any EPIC-like metric $d$, but where $R_1$ and $R_2$ induce the same ordering of policies for any choice of transition function and any choice of initial state distribution.*

*Proof.* Recall that $\mathcal{S}$ must contain at least two states $s_1, s_2$, and $\mathcal{A}$ must contain at least two actions $a_1, a_2$. Let $R_1(s_1, a_1, s_1) = 1$, $R_1(s_1, a_1, s_2) = \epsilon$, $R_2(s_1, a_1, s_1) = \epsilon$, and $R_2(s_1, a_1, s_2) = 1$, and let $R_1$ and $R_2$ be 0 for all other transitions. $R_1$ and $R_2$ do not differ by potential shaping or positive linear scaling; this means that $d(R_1, R_2) > 0$ for any EPIC-like metric $d$. However, $R_1$ and $R_2$ have the same policy ordering for all $\tau$ and $\mu_0$. $\square$

### F.4 Other Proofs

In this section, we provide a the remaining proofs of the results mentioned in the main text.

**Proposition 1.** *Any STARC metric is a pseudometric on $\mathcal{R}$.*

*Proof.* To show that $d$ is a pseudometric, we must show that

1. $d(R, R) = 0$

2. $d(R_1, R_2) = d(R_2, R_1)$

3. $d(R_1, R_3) \leqslant d(R_1, R_2) + d(R_2, R_3)$

1 follows from the fact that $m$ is a metric, and 2 follows directly from the fact that the definition of STARC metrics is symmetric in $R_1$ and $R_2$. For 3, the fact that $m$ is a metric again implies that $d(R_1, R_3) = m(s(R_1), s(R_3)) \leqslant m(s(R_1), s(R_2)) + m(s(R_2), s(R_3)) = d(R_1, R_2) + d(R_2, R_3)$. This completes the proof. $\qquad\square$

**Proposition 2.** *All STARC metrics have the property that* $d(R_1, R_2) = 0$ *if and only if* $R_1$ *and* $R_2$ *induce the same ordering of policies.*

*Proof.* This is immediate from Proposition 1, together with the fact that if $R_1$ and $R_2$ differ by potential shaping, $S'$-redistribution, and positive linear scaling, applied in any order, then $R_2 = \alpha \cdot R_3$ for some scalar $\alpha$ and some $R_3$ that differs from $R_1$ via potential shaping and $S'$-redistribution. $\quad\square$

**Proposition 3.** *If two pseudometrics* $d_1$, $d_2$ *on* $\mathcal{R}$ *are both sound and complete, then* $d_1$ *and* $d_2$ *are bilipschitz equivalent.*

*Proof.* Since $d_1$ is complete, we have that

$$L_1 \cdot d_1(R_1, R_2) \cdot (\max_\pi J_1(\pi) - \min_\pi J_1(\pi)) \leqslant \max_{\pi_1, \pi_2 : J_2(\pi_2) \geqslant J_2(\pi_1)} J_1(\pi_1) - J_1(\pi_2).$$

Similarly, since $d_2$ is sound, we also have that

$$\max_{\pi_1, \pi_2 : J_2(\pi_2) \geqslant J_2(\pi_1)} J_1(\pi_1) - J_1(\pi_2) \leqslant U_2 \cdot d_2(R_1, R_2) \cdot (\max_\pi J_1(\pi) - \min_\pi J_1(\pi)).$$

This implies that

$$L_1 \cdot d_1(R_1, R_2) \cdot (\max_\pi J_1(\pi) - \min_\pi J_1(\pi)) \leqslant U_2 \cdot d_2(R_1, R_2) \cdot (\max_\pi J_1(\pi) - \min_\pi J_1(\pi)).$$

First suppose that $(\max_\pi J_1(\pi) - \min_\pi J_1(\pi)) > 0$. We can then divide both sides, and obtain that

$$d_1(R_1, R_2) \leqslant \left(\frac{U_2}{L_1}\right) d_2(R_1, R_2).$$

Similarly, we also have that

$$\left(\frac{L_2}{U_1}\right) d_2(R_1, R_2) \leqslant d_1(R_1, R_2).$$

This means that we have constants $\left(\frac{U_2}{L_1}\right)$ and $\left(\frac{L_2}{U_1}\right)$ not depending on $R_1$ or $R_2$, such that

$$\left(\frac{L_2}{U_1}\right) d_2(R_1, R_2) \leqslant d_1(R_1, R_2) \leqslant \left(\frac{U_2}{L_1}\right) d_2(R_1, R_2)$$

for all $R_1$ and $R_2$ such that $(\max_\pi J_1(\pi) - \min_\pi J_1(\pi)) > 0$.

Next, assume $(\max_\pi J_1(\pi) - \min_\pi J_1(\pi)) = 0$ but $(\max_\pi J_2(\pi) - \min_\pi J_2(\pi)) > 0$. Since $d_1$ and $d_2$ are pseudometrics, we have that $d_1(R_1, R_2) = d_1(R_2, R_1)$ and $d_2(R_1, R_2) = d_2(R_2, R_1)$. Therefore, $\left(\frac{L_2}{U_1}\right) d_2(R_1, R_2) \leqslant d_1(R_1, R_2) \leqslant \left(\frac{U_2}{L_1}\right) d_2(R_1, R_2)$ in this case as well.

Finally, assume that $(\max_\pi J_1(\pi) - \min_\pi J_1(\pi)) = 0$ and $(\max_\pi J_2(\pi) - \min_\pi J_2(\pi)) = 0$. In this case, $R_1$ and $R_2$ induce the same policy order (namely, the order where $\pi_1 \equiv \pi_2$ for all $\pi_1, \pi_2$). This in turn means that $d_1(R_1, R_2) = d_2(R_1, R_2) = 0$, and so $\left(\frac{L_2}{U_1}\right) d_2(R_1, R_2) \leqslant d_1(R_1, R_2) \leqslant \left(\frac{U_2}{L_1}\right) d_2(R_1, R_2)$ in this case as well. This completes the proof. $\qquad\square$

## G  EXPERIMENTAL SETUP OF SMALL MDPS

In this appendix, we give the precise details required to reproduce our experimental results, as well as more of the raw data than what is provided in the main text.

### G.1 Environments and Rewards

As mentioned in the main text, we used Markov Decision Processes with 32 states and 4 actions. The discount factor was set to 0.95 and the initial state distribution was uniform.

The transition distribution $\tau(s, a, s')$ was generated as follows:

1. Sample i.i.d. Gaussians ($\mu = 0, \sigma = 1$) to generate a matrix of shape $[32, 4, 32]$.

2. For each item in the matrix: if the item is below 1, set its value to -20. This is done to ensure the transition distribution is sparse and therefore more similar to real-world environments. Without this step, when an agent is in state $S$ and takes action $A$, the distribution $\tau(S, A, s')$ would be close to uniform, meaning that the choice of the action would not make much of a difference.

3. Softmax along the last dimension (which corresponds to $s'$) to get a valid probability distribution.

We then generated pairs of rewards. This worked in two stages: random generation and interpolation.

In the random generation stage, we choose two random rewards $R_1, R_2$ using the following procedure:

1. Sample i.i.d. Gaussians ($\mu = 0, \sigma = 1$) to generate a matrix of shape $[32, 4, 32]$ corresponding to $R(s, a, s')$.

2. With a 20% probability, make the function sparse in the following way – for each item in the matrix: if the item is below 3, set its value to 0.

3. With a 70% probability, scale the reward function in the following way – sample a uniform distribution between 0 and 10, multiply the matrix by this number.

4. With a 30% probability, translate the reward function in the following way – sample a uniform distribution between 0 and 10, add this number to the matrix.

5. With a 50% probability, apply random potential shaping in the following way – sample 32 i.i.d. Gaussians ($\mu = 0, \sigma = 1$) to get a potential vector $\Phi$. Then sample a uniform distribution between 0 and 10 and multiply the vector by this number. Then sample a uniform distribution between 0 and 1 and add this number to the vector. Then apply potential shaping to the reward function: $R_{\text{new}}(s, a, s') = R(s, a, s') + \gamma\Phi(s') - \Phi(s)$.

When we say "With an X% probability", we are sampling a random number from a uniform distribution between 0 and 1 and if the number is above (100-X)%, we perform the action described, otherwise we skip the step.

Then in the interpolation stage, we take the pair of reward functions generated above and do a linear interpolation between them, finding 16 functions which lie between $R_1$ and $R_2$. More precisely, we set $R_{(i)} = R_1 + id$ where $d = (R_2 - R_1)/16$ and with $i$ ranging from 1 to 16.

The interpolation step exists to give us pairs of rewards which are relatively close to each other – for instance, $R_1$ and $R_{(1)}$ are very similar. This is important because nearly all reward functions generated with the random generation process described above will be orthogonal to each other, and thus their distances to each other would always be quite large. By including this interpolation step, we ensure a greater variety in the range of distance values and regret values we expect to see.

For each environment, we generated 16 pairs of reward functions, and then for each pair we performed 16 interpolation steps. This means that for each transition distribution, we compared 256 different reward functions.

We then compute all distance metrics as well as rollout regret between $R_1$ and $R_{(i)}$ for all $i$.

### G.2 Rollout Regret

We calculate rollout regret in 3 stages: 1, find optimal and anti-optimal policies, 2, compute returns under various policies and reward functions, 3, calculate regret.

In stage 1, we use value iteration to find policies $\pi_1$ which maximises reward $R_1$, $\pi_{(i)}$ which maximises reward $R_{(i)}$, $\pi_x$ which minimises reward $R_1$ (in other words maximising reward $-R_1$,

meaning $\pi_x$ is the worst possible policy under $R_1$), and $\pi_y$ which minimises reward $R_2$. Note that all of these policies are deterministic, i.e. $\pi : \mathcal{S} \to \mathcal{A}$.

In stage 2, we simulate a number of episodes to determine the average return of the policy. Specifically, we simulate 32 episodes such that no two episodes start in the same initial state (this helps reduce noise in the return estimates). The episode terminates when the discount factor being applied (i.e. $\gamma^t$) is below $10^{-5}$.

In stage 3, we calculate regret as follows. The regret is the average of two regrets: the regret of using $\pi_{(i)}$ instead of $\pi_1$ when evaluating using $R_1$, and the regret of using $\pi_1$ instead of $\pi_{(i)}$ when evaluating using $R_{(i)}$ (with both of these being normalised by the range of possible returns):

$$\text{Reg} = \frac{\text{Reg}_1 + \text{Reg}_{(i)}}{2}$$

$$\text{Reg}_1 = \frac{J_1(\pi_1) - J_1(\pi_{(i)})}{J_1(\pi_1) - J_1(\pi_x)}$$

$$\text{Reg}_{(i)} = \frac{J_{(i)}(\pi_{(i)}) - J_{(i)}(\pi_1)}{J_{(i)}(\pi_{(i)}) - J_{(i)}(\pi_y)}$$

In cases where the denominator is zero, we simply replace it with 1 (since the numerator in these cases is also necessarily 0).

### G.3 LIST OF METRICS

Our experiment covers hundreds of metrics, derived by creating different combinations of canonicalisation functions, normalisations, and distance metrics. Specifically, we used 6 "pseudo-canonicalisations" (some of which, like $C^{\text{EPIC}}$ and $C^{\text{DARD}}$, do not meet the conditions of Definition 1), 7 normalisation functions, and 6 distance norms.

For canonicalisations, we used `None` (which simply skips the canonicalisation step), $C^{\text{EPIC}}$, $C^{\text{DARD}}$, `MinimalPotential` (which is the minimal "pseudo-canonicalisation" that removes potential shaping but not $S'$-redistribution, and therefore is easier to compute), `VALPotential` (which is given by $R(s, a, s') - V^\pi(s) + \gamma V^\pi(s')$), and `VAL` (defined in Proposition 2 as $\mathbb{E}_{S' \sim \tau(s,a)}[R(s, a, S') - V^\pi(s) + \gamma V^\pi(S')]$). For both $C^{\text{EPIC}}$ and $C^{\text{DARD}}$, both $\mathcal{D}_\mathcal{S}$ and $\mathcal{D}_\mathcal{A}$ were chosen to be uniform over $\mathcal{S}$ and $\mathcal{A}$. For both `VALPotential` and `VAL`, $\pi$ was chosen to be the uniformly random policy. [9] Note that `VAL` is the only canonicalisation which removes both potential shaping and $S'$-redistribution, and thus the only one that meets the STARC definition of a canonicalisation function (Definition 1). The other pseudo-canonicalisations were used for comparison. It is worth noting that our experiment does *not* include the minimal canonicalisation functions, given in Definition 4, because these functions are prohibitively expensive to compute. They are therefore better suited for theoretical analysis, rather than practical evaluations.

The normalisation step and the distance step used $L_1$, $L_2$, $L_\infty$, `weighted_`$L_1$, `weighted_`$L_2$, and `weighted_`$L_\infty$. The weighted norms are weighted by the transition function $\tau$, i.e. $L_p^\tau(R)(s, a, s') = (\sum_{s,a,s'} \tau(s, a, s')|R(s, a, s')|^p)^{1/p}$. We also considered metrics that skip the normalisation step.

We used almost all combinations of these – the only exception was that we did not combine `MinimalPotential` with normalisation norms $L_\infty$ or `weighted_`$L_\infty$ because the optimisation algorithm for `MinimalPotential` does not converge for these norms.

### G.4 NUMBER OF REWARD PAIRS

We used 49,152 reward pairs. This number was chosen in advance as the stopping point – it corresponds to using 96 CPU cores, generating 2 environments on each core, choosing 16 reward pairs within each environment and then performing 16 interpolation steps between them.

---

[9]Since `VAL` is a valid canonicalisation function for any choice of policy $\pi$, we simply picked a policy for which $V^\pi$ would be easy to estimate. The reason for choosing a uniformly random policy, rather than some deterministic policy, is that this policy has exploration build in.

# H   FULL RESULTS OF SMALL MDP EXPERIMENTS

We used the Balrog GPU cluster at UC Berkeley, which consists of 8 A100 GPUs, each with 40 GB memory, along with 96 CPU cores.

The notation in this table is in the format `Canonicalisation-normalisation-distance`. For instance, `VAL-2-weighted_1` means using the `Val` canonicalisation function, $L_2$ normalisation function, and then taking the distance with the weighted $L_1$ norm. `0` means normalisation is skipped.

Table 2: Full experimental results

| Distance function | Correlation to regret |
|---|---|
| VALPotential-1-weighted_1 | 0.876 |
| VAL-1-weighted_1 | 0.873 |
| VAL-1-1 | 0.873 |
| VAL-weighted_1-weighted_1 | 0.873 |
| VAL-weighted_1-1 | 0.873 |
| VAL-1-2 | 0.870 |
| VAL-weighted_1-2 | 0.870 |
| VAL-1-weighted_2 | 0.870 |
| VAL-weighted_1-weighted_2 | 0.870 |
| VALPotential-weighted_1-weighted_1 | 0.867 |
| DARD-1-weighted_1 | 0.861 |
| VAL-weighted_2-inf | 0.858 |
| VAL-2-inf | 0.858 |
| VAL-weighted_2-weighted_2 | 0.856 |
| VAL-2-2 | 0.856 |
| VAL-weighted_2-2 | 0.856 |
| VAL-2-weighted_2 | 0.856 |
| VALPotential-weighted_2-weighted_2 | 0.845 |
| DARD-weighted_1-weighted_1 | 0.835 |
| DARD-weighted_2-weighted_2 | 0.831 |
| EPIC-weighted_1-weighted_1 | 0.830 |
| VALPotential-2-weighted_2 | 0.828 |
| DARD-2-weighted_2 | 0.826 |
| DARD-1-1 | 0.824 |
| EPIC-weighted_2-weighted_2 | 0.823 |
| EPIC-1-weighted_1 | 0.819 |
| VAL-weighted_inf-inf | 0.816 |
| MinimalPotential-1-1 | 0.815 |
| MinimalPotential-2-weighted_2 | 0.814 |
| EPIC-2-weighted_2 | 0.814 |
| EPIC-1-1 | 0.814 |
| VALPotential-weighted_2-weighted_inf | 0.807 |
| EPIC-weighted_2-weighted_inf | 0.806 |
| VAL-2-1 | 0.804 |
| VAL-2-weighted_1 | 0.804 |
| VAL-weighted_2-1 | 0.804 |
| VAL-weighted_2-weighted_1 | 0.804 |
| VALPotential-1-1 | 0.800 |
| VALPotential-weighted_2-weighted_1 | 0.784 |
| VAL-weighted_2-weighted_inf | 0.783 |
| VAL-2-weighted_inf | 0.783 |
| VALPotential-2-2 | 0.782 |
| DARD-2-2 | 0.782 |
| MinimalPotential-2-2 | 0.778 |
| EPIC-2-2 | 0.778 |
| VALPotential-1-weighted_2 | 0.776 |

| | |
|---|---|
| DARD-weighted_2-weighted_1 | 0.774 |
| DARD-2-weighted_1 | 0.767 |
| VALPotential-2-weighted_1 | 0.767 |
| VAL-weighted_inf-weighted_2 | 0.766 |
| VAL-weighted_inf-2 | 0.766 |
| DARD-1-weighted_2 | 0.761 |
| VAL-inf-inf | 0.756 |
| DARD-weighted_1-weighted_2 | 0.754 |
| VALPotential-2-1 | 0.752 |
| DARD-2-1 | 0.751 |
| EPIC-weighted_1-1 | 0.749 |
| VAL-1-inf | 0.749 |
| VAL-weighted_1-inf | 0.749 |
| MinimalPotential-2-weighted_1 | 0.746 |
| EPIC-2-weighted_1 | 0.746 |
| VAL-1-weighted_inf | 0.741 |
| VAL-weighted_1-weighted_inf | 0.741 |
| EPIC-weighted_2-weighted_1 | 0.738 |
| VALPotential-weighted_inf-weighted_inf | 0.735 |
| EPIC-2-1 | 0.734 |
| VAL-weighted_inf-weighted_inf | 0.734 |
| MinimalPotential-2-1 | 0.733 |
| EPIC-weighted_1-weighted_2 | 0.730 |
| VAL-inf-weighted_2 | 0.723 |
| VAL-inf-2 | 0.723 |
| VALPotential-weighted_inf-weighted_1 | 0.722 |
| MinimalPotential-1-weighted_1 | 0.718 |
| DARD-weighted_inf-weighted_2 | 0.718 |
| DARD-weighted_inf-weighted_1 | 0.713 |
| DARD-weighted_2-1 | 0.711 |
| DARD-weighted_2-weighted_inf | 0.708 |
| VAL-inf-weighted_1 | 0.708 |
| VAL-inf-1 | 0.708 |
| VAL-inf-weighted_inf | 0.707 |
| VAL-weighted_inf-1 | 0.707 |
| VAL-weighted_inf-weighted_1 | 0.707 |
| DARD-weighted_inf-weighted_inf | 0.698 |
| VALPotential-weighted_1-weighted_inf | 0.692 |
| EPIC-weighted_inf-weighted_2 | 0.692 |
| EPIC-weighted_inf-weighted_1 | 0.686 |
| VALPotential-1-weighted_inf | 0.685 |
| DARD-weighted_inf-1 | 0.685 |
| EPIC-weighted_2-1 | 0.680 |
| DARD-weighted_1-weighted_inf | 0.679 |
| VALPotential-2-weighted_inf | 0.677 |
| DARD-2-weighted_inf | 0.675 |
| EPIC-weighted_inf-weighted_inf | 0.661 |
| DARD-inf-weighted_1 | 0.657 |
| DARD-1-weighted_inf | 0.654 |
| VALPotential-inf-weighted_1 | 0.653 |
| DARD-inf-weighted_2 | 0.652 |
| VALPotential-inf-weighted_2 | 0.648 |
| EPIC-1-2 | 0.647 |
| EPIC-weighted_inf-1 | 0.642 |
| DARD-inf-1 | 0.639 |
| DARD-inf-2 | 0.637 |
| MinimalPotential-2-weighted_inf | 0.637 |
| EPIC-2-weighted_inf | 0.637 |
| VALPotential-inf-1 | 0.636 |

| | |
|---|---|
| VALPotential-inf-2 | 0.634 |
| MinimalPotential-2-inf | 0.634 |
| EPIC-2-inf | 0.634 |
| None-2-weighted_2 | 0.633 |
| DARD-inf-weighted_inf | 0.632 |
| DARD-2-inf | 0.630 |
| EPIC-inf-weighted_2 | 0.630 |
| EPIC-inf-weighted_1 | 0.629 |
| VALPotential-2-inf | 0.625 |
| VALPotential-inf-weighted_inf | 0.624 |
| EPIC-1-weighted_2 | 0.622 |
| None-weighted_2-weighted_inf | 0.622 |
| DARD-weighted_1-1 | 0.621 |
| EPIC-inf-1 | 0.620 |
| None-2-2 | 0.618 |
| EPIC-inf-2 | 0.617 |
| None-weighted_2-weighted_2 | 0.615 |
| EPIC-weighted_1-weighted_inf | 0.607 |
| None-weighted_1-weighted_1 | 0.598 |
| None-1-1 | 0.597 |
| None-2-weighted_1 | 0.579 |
| MinimalPotential-1-2 | 0.576 |
| None-weighted_2-weighted_1 | 0.573 |
| None-2-1 | 0.571 |
| EPIC-inf-weighted_inf | 0.571 |
| MinimalPotential-1-weighted_2 | 0.568 |
| VALPotential-inf-inf | 0.557 |
| None-inf-inf | 0.555 |
| EPIC-inf-inf | 0.554 |
| DARD-inf-inf | 0.552 |
| EPIC-1-inf | 0.542 |
| None-inf-weighted_2 | 0.539 |
| None-weighted_inf-weighted_1 | 0.539 |
| None-inf-2 | 0.538 |
| None-inf-weighted_1 | 0.537 |
| None-inf-1 | 0.537 |
| None-weighted_inf-weighted_inf | 0.530 |
| EPIC-weighted_1-2 | 0.529 |
| EPIC-1-weighted_inf | 0.517 |
| None-1-weighted_1 | 0.515 |
| None-2-weighted_inf | 0.513 |
| MinimalPotential-1-inf | 0.493 |
| MinimalPotential-1-weighted_inf | 0.489 |
| None-inf-weighted_inf | 0.487 |
| DARD-1-2 | 0.453 |
| None-2-inf | 0.444 |
| EPIC-weighted_1-inf | 0.436 |
| None-weighted_1-weighted_inf | 0.429 |
| None-1-weighted_2 | 0.390 |
| DARD-1-inf | 0.376 |
| None-1-weighted_inf | 0.353 |
| None-1-2 | 0.352 |
| None-0-inf | 0.333 |
| DARD-weighted_inf-2 | 0.319 |
| None-0-weighted_inf | 0.312 |
| None-0-2 | 0.304 |
| None-0-weighted_2 | 0.303 |
| None-0-weighted_1 | 0.296 |
| None-0-1 | 0.296 |

| | |
|---|---|
| None-1-inf | 0.278 |
| DARD-weighted_1-2 | 0.241 |
| VALPotential-1-2 | 0.240 |
| EPIC-weighted_2-2 | 0.224 |
| DARD-weighted_1-inf | 0.220 |
| EPIC-weighted_inf-2 | 0.197 |
| EPIC-0-inf | 0.184 |
| DARD-weighted_2-2 | 0.178 |
| VALPotential-1-inf | 0.171 |
| VAL-0-weighted_inf | 0.171 |
| VALPotential-0-inf | 0.171 |
| DARD-0-inf | 0.170 |
| VAL-0-weighted_1 | 0.149 |
| VAL-0-1 | 0.149 |
| VAL-0-weighted_2 | 0.146 |
| VAL-0-2 | 0.146 |
| VALPotential-0-weighted_inf | 0.141 |
| DARD-0-weighted_inf | 0.141 |
| VAL-0-inf | 0.140 |
| EPIC-0-weighted_inf | 0.140 |
| DARD-0-weighted_1 | 0.137 |
| VALPotential-0-weighted_1 | 0.136 |
| DARD-0-weighted_2 | 0.136 |
| VALPotential-0-weighted_2 | 0.135 |
| EPIC-0-weighted_2 | 0.131 |
| EPIC-0-weighted_1 | 0.130 |
| EPIC-weighted_2-inf | 0.129 |
| DARD-0-2 | 0.125 |
| DARD-0-1 | 0.124 |
| VALPotential-0-2 | 0.123 |
| DARD-weighted_2-inf | 0.122 |
| VALPotential-0-1 | 0.122 |
| EPIC-0-1 | 0.122 |
| EPIC-0-2 | 0.122 |
| VALPotential-weighted_2-1 | 0.112 |
| DARD-weighted_inf-inf | 0.095 |
| VALPotential-weighted_2-2 | 0.093 |
| VALPotential-weighted_inf-1 | 0.077 |
| VALPotential-weighted_inf-weighted_2 | 0.073 |
| VALPotential-weighted_2-inf | 0.065 |
| EPIC-weighted_inf-inf | 0.052 |
| VALPotential-weighted_inf-2 | 0.051 |
| VALPotential-weighted_inf-inf | 0.024 |
| None-weighted_2-1 | -0.034 |
| VALPotential-weighted_1-1 | -0.035 |
| None-weighted_inf-1 | -0.035 |
| VALPotential-weighted_1-weighted_2 | -0.037 |
| None-weighted_inf-weighted_2 | -0.040 |
| VALPotential-weighted_1-2 | -0.040 |
| None-weighted_inf-2 | -0.043 |
| None-weighted_2-2 | -0.043 |
| VALPotential-weighted_1-inf | -0.044 |
| None-weighted_1-1 | -0.045 |
| None-weighted_inf-inf | -0.046 |
| None-weighted_2-inf | -0.046 |
| None-weighted_1-weighted_2 | -0.047 |
| None-weighted_1-2 | -0.047 |
| None-weighted_1-inf | -0.048 |

## H.1 COMPARISON OF EXPERIMENTAL PERFORMANCE BASED ON CHOICE OF NORMS

As discussed in the main text, the choice of normalisation and metric functions can make a noticeable difference to the performance of a reward metric. To make it easier to see the impact that this choice has, this appendix contains the same data as Appendix H, but organised together by canonicalisation function, and then arranged by normalisation and metric.

|  | 1 | 2 | inf | weighted_1 | weighted_2 | weighted_inf |
|---|---|---|---|---|---|---|
| 0 | 0.296 | 0.304 | 0.333 | 0.296 | 0.303 | 0.312 |
| 1 | 0.597 | 0.352 | 0.278 | 0.515 | 0.39 | 0.353 |
| 2 | 0.571 | 0.618 | 0.444 | 0.579 | **0.633** | 0.513 |
| inf | 0.537 | 0.538 | 0.555 | 0.537 | 0.539 | 0.487 |
| weighted_1 | -0.045 | -0.047 | -0.048 | 0.598 | -0.047 | 0.429 |
| weighted_2 | -0.034 | -0.043 | -0.046 | 0.573 | 0.615 | 0.622 |
| weighted_inf | -0.035 | -0.043 | -0.046 | 0.539 | -0.04 | 0.53 |

Table 3: Correlation to regret for the `None` canonicalisation for each normalization and distance metric. Each row corresponds to a normalisation function, and each column corresponds to a metric function.

|  | 1 | 2 | inf | weighted_1 | weighted_2 | weighted_inf |
|---|---|---|---|---|---|---|
| 0 | 0.122 | 0.122 | 0.184 | 0.13 | 0.131 | 0.14 |
| 1 | 0.814 | 0.647 | 0.542 | 0.819 | 0.622 | 0.517 |
| 2 | 0.734 | 0.778 | 0.634 | 0.746 | 0.814 | 0.637 |
| inf | 0.62 | 0.617 | 0.554 | 0.629 | 0.63 | 0.571 |
| weighted_1 | 0.749 | 0.529 | 0.436 | **0.83** | 0.73 | 0.607 |
| weighted_2 | 0.68 | 0.224 | 0.129 | 0.738 | 0.823 | 0.806 |
| weighted_inf | 0.642 | 0.197 | 0.052 | 0.686 | 0.692 | 0.661 |

Table 4: Correlation to regret for the `EPIC` canonicalisation for each normalization and distance metric. Each row corresponds to a normalisation function, and each column corresponds to a metric function.

|  | 1 | 2 | inf | weighted_1 | weighted_2 | weighted_inf |
|---|---|---|---|---|---|---|
| 0 | 0.124 | 0.125 | 0.17 | 0.137 | 0.136 | 0.141 |
| 1 | 0.824 | 0.453 | 0.376 | **0.861** | 0.761 | 0.654 |
| 2 | 0.751 | 0.782 | 0.63 | 0.767 | 0.826 | 0.675 |
| inf | 0.639 | 0.637 | 0.552 | 0.657 | 0.652 | 0.632 |
| weighted_1 | 0.621 | 0.241 | 0.22 | 0.835 | 0.754 | 0.679 |
| weighted_2 | 0.711 | 0.178 | 0.122 | 0.774 | 0.831 | 0.708 |
| weighted_inf | 0.685 | 0.319 | 0.095 | 0.713 | 0.718 | 0.698 |

Table 5: Correlation to regret for the `DARD` canonicalisation for each normalization and distance metric. Each row corresponds to a normalisation function, and each column corresponds to a metric function.

|  | 1 | 2 | inf | weighted_1 | weighted_2 | weighted_inf |
|---|---|---|---|---|---|---|
| 1 | **0.815** | 0.576 | 0.493 | 0.718 | 0.568 | 0.489 |
| 2 | 0.733 | 0.778 | 0.634 | 0.746 | 0.814 | 0.637 |

Table 6: Correlation to regret for the `MinimalPotential` canonicalisation for each normalization and distance metric. Each row corresponds to a normalisation function, and each column corresponds to a metric function.

|  | 1 | 2 | inf | weighted_1 | weighted_2 | weighted_inf |
|---|---|---|---|---|---|---|
| 0 | 0.122 | 0.123 | 0.171 | 0.136 | 0.135 | 0.141 |
| 1 | 0.8 | 0.24 | 0.171 | **0.876** | 0.776 | 0.685 |
| 2 | 0.752 | 0.782 | 0.625 | 0.767 | 0.828 | 0.677 |
| inf | 0.636 | 0.634 | 0.557 | 0.653 | 0.648 | 0.624 |
| weighted_1 | -0.035 | -0.04 | -0.044 | 0.867 | -0.037 | 0.692 |
| weighted_2 | 0.112 | 0.093 | 0.065 | 0.784 | 0.845 | 0.807 |
| weighted_inf | 0.077 | 0.051 | 0.024 | 0.722 | 0.073 | 0.735 |

Table 7: Correlation to regret for the `VALPotential` canonicalisation for each normalization and distance metric. Each row corresponds to a normalisation function, and each column corresponds to a metric function.

|  | 1 | 2 | inf | weighted_1 | weighted_2 | weighted_inf |
|---|---|---|---|---|---|---|
| 0 | 0.149 | 0.146 | 0.14 | 0.149 | 0.146 | 0.171 |
| 1 | **0.873** | 0.87 | 0.749 | **0.873** | 0.87 | 0.741 |
| 2 | 0.804 | 0.856 | 0.858 | 0.804 | 0.856 | 0.783 |
| inf | 0.708 | 0.723 | 0.756 | 0.708 | 0.723 | 0.707 |
| weighted_1 | **0.873** | 0.87 | 0.749 | **0.873** | 0.87 | 0.741 |
| weighted_2 | 0.804 | 0.856 | 0.858 | 0.804 | 0.856 | 0.783 |
| weighted_inf | 0.707 | 0.766 | 0.816 | 0.707 | 0.766 | 0.734 |

Table 8: Correlation to regret for the `VAL` canonicalisation for each normalization and distance metric. Each row corresponds to a normalisation function, and each column corresponds to a metric function.

# I EXPERIMENTAL SETUP OF REACHER ENVIRONMENT

In this appendix, we elaborate on the details of our Reacher experiments. The discount rate for this experiment was $\gamma = 0.99$, following the original MuJoCo environment.

## I.1 REWARD FUNCTIONS

As mentioned in the main text, we used 7 different reward functions.

The `GroundTruth` is simply copied from the original Reacher environment. It computes the Euclidean distance between the fingertip and target, denoted $d$. It also computes a penalty for taking large actions, which is computed by squaring the values of the action and summing them, $p = a_0^2 + a_1^2$. It then returns $-(d + p)$.

The `PotentialShaped` reward applies randomly generated (but deterministic) potential shaping on top of `GroundTruth`. When the experiment starts, 11 weights (one for each dimension in observation space) and 1 bias are randomly sampled from a normal Gaussian. The potential function is then simply $\Phi(s) = w \cdot x + b$, so the full reward is `PotentialShaped`$(s, a, s') = $ `GroundTruth`$(s, a, s') + \gamma\Phi(s') - \Phi(s)$.

`SPrime` returns the same value as `GroundTruth` if $s' = \tau(s, a)$, and the same value as `Random` otherwise. At the start of the experiment, a new instance of `Random` is initialised (see below for details). We consider $s'$ to follow from $\tau(s, a)$ if the two quantities are either less than 1% apart, or less than 0.01 apart (even if they aren't perfectly equal).

`SecondPeak` creates a second, smaller "peak" (corresponding to a second, less important target) in the environment, alongside the original "peak" from `GroundTruth`. When initialised at the start of the experiment, it picks a random position on the same 2D plane where the fingertip and target are, such that the Euclidean distance between the second peak and the original peak is at least 0.5 (note that the size of the whole plane is $1 \times 1$). The reward is then determined by first computing the Euclidean distance between the fingertip and the second peak, denoted $d$, and then simply adding $-0.2d$ on top of `GroundTruth`, ie. `SecondPeak` = `GroundTruth` $- 0.2d$.

`SemanticallyIdentical` creates a reward peak around the target, similarly to `GroundTruth`, but this peak has a different shape. It is a 2D Gaussian with a standard deviation of 0.1 along both axes. The values of the Gaussian are then rounded to the nearest 0.01.

`NegativeGroundReward` simply returns $-1 * $ `GroundTruth`.

`Random` returns random (but deterministic) values. When the experiment starts, 11 $s$-weights, 2 $a$-weights, 11 $s'$-weights, and 1 bias are generated by sampling a normal Gaussian. The reward function then simply returns $s \cdot w_s + a \cdot w_a + s' \cdot w_{s'} + b$.

## I.2 CANONICALISING AND NORMALISING IN CONTINUOUS SETTINGS

The state value function in VAL is based on a uniformly random policy $\pi$. We implemented $V^\pi$ using SARSA (Rummery & Niranjan, 1994) updates with AdamW (Loshchilov & Hutter, 2019) and a reply buffer on a 4-layer MLP (which maps observations onto real values).

For the norm, we effectively need to compute the norm of a function, which means taking the norm of an infinite-dimensional vector. This can be written precisely as $(\int |f(x)|^p \, dx)^{1/p}$, and approximated using Monte Carlo sampling as $(\frac{1}{N} \sum |f(x)|^p \, dx)^{1/p}$. When taking a sample of the reward function, we sample $s, a$ uniformly, and then set $s' = \tau(s, a)$ – this removes impossible transitions from the sample space, while also reducing the dimensionality of the space we need to cover from 22 dimensions down to 12. As a special case, when taking the $L_\infty$ norm, we simply look for the maximum value of $|f(x)|$. This could be approximated using optimisation algorithms by assuming $|f(x)|$ is convex, but we chose not to make this assumption and instead simply choose the maximum among the samples.

