# OpenReview forum: "STARC: A General Framework For Quantifying Differences Between Reward Functions"
_ICLR.cc/2024/Conference — ICLR 2024 poster_

### Official Review · Reviewer_Sb1v · 2023-10-14

**Soundness:** 4 excellent
**Presentation:** 3 good
**Contribution:** 3 good
**Rating:** 8
**Confidence:** 4

**Summary:**

This paper proposes a new family of metrics to measure similarity between reward functions in the MDP setting. The proposed family is based on a strong definition of regret and is defined such that it allows to estimate both an upper and a lower bound on such difference, so that similar functions are guaranteed to also yield a similar policy ordering. Also, the mathematical definition is simple and can well be applied both theoretically and empirically.

**Strengths:**

This work investigates a very interesting problem, as it is often unclear how and to what extent different reward functions can lead to the same policy preferences, and in general how different rewards can be compared. The theoretical characterization is rigorous but also easily readable, which is not always obvious in this kind of math-heavy works.

**Weaknesses:**

I do not have that much to say about eventual weaknesses, other than some points about the empirical results presentation (detailed in the Questions below).

**Questions:**

- The definition of a pseudometric by a norm seems wrong: why $m(x,y)=n(x-v)$? What is $v$? Should not that be $n(x-y)$?
- The results in Figure 1 are a bit difficult to parse. Each column corresponds to a different metric family, and each point is a different choice of either the metric $m$, the norm $n$ or the canonicalisation $c$. But what is the None column? Are the choices of $m$, $n$ and $c$ consistent throughout different families? Although I appreciate the space constraints imposed by the venue, I really feel that some details on how these are chosen should fit into this section, as these are crucial to properly assess the proposed results.
- I think it would have been also interesting to assess the individual impact of the choice of $m$, $n$ or $c$ when the values of the other is fixed: for example, how important is the choice of the canonicalisation $c$ w.r.t. the chosen metric and norm? Although there is some discussion w.r.t. this at the end of page 8, it is a bit difficult for me to relate such discussion to the results in Figure 1.

---

> ### Author Response · Authors · 2023-11-13
> **Response**
>
> We thank reviewer Sb1v for their review! Our answers to your questions are as follows:
>
> Questions:
> 1. You are right, it should read $n(|x-y|)$. We will fix this in the text.
> 2. The None column corresponds to the case where no canonicalisation function is used. The choices of $m$ and $n$ are consistent throughout these families, except that the MinimalPotential canonicalisation function was not used with the $L_\infty$-norms, because the optimisation algorithm for MinimalPotential fails to converge for these norms. The full details are provided in Appendix F (and especially F.3), but we will move more of this information into the main text, to make the figure easier to parse.
> 3. We agree that this comparison is relevant, though it would be difficult to fit it in Section 4 given the space constraints. This information *can* be read from the raw data provided in Appendix G, but we will make this information more accessible.
>
> We hope that this clarifies your questions!

---

> > ### Comment · Reviewer_Sb1v · 2023-11-14
> > **Reply to authors**
> >
> > I thank the authors for their replies, which have proved useful in answering my questions. Some additional comments:
> >
> > 2. This is clear now, but it probably needs to be made explicit and clear in the paper as well, as that Figure is the core empirical result of the entire work.
> > 3. If Appendix G is already reporting raw data for this, I suggest to turn these into a proper comparison (that can be left in the appendix, as I perfectly understand how the space limitations are not allowing everything to make it into the main body of the work).

---

### Official Review · Reviewer_JNkN · 2023-10-27

**Soundness:** 3 good
**Presentation:** 3 good
**Contribution:** 3 good
**Rating:** 6
**Confidence:** 3

**Summary:**

Considering that there is a lack of good methods for quantifying the difference between reward functions, this paper proposes STARC, a class of pseudo metrics on the space of all reward functions. Authors show that STARC metrics are tight, which gives both an upper and a lower bound on worst-case regret. Empirical results demonstrate the practical efficacy of STARC metrics.

**Strengths:**

1. This paper is clearly written, with extensive theoretical results and empirical verification.
2. Quantifying the differences of reward functions, especially from the perspective of their induced orders of policies is an interesting and important topic in RL, but is lack of consideration. This paper makes a new step.

**Weaknesses:**

1. From the experimental results shown in Figure 1, we can see that the choice of normalization function $n$ and distance metric $m$ can have a significant impact on the metric’s accuracy. But the theoretical results presented in this paper cannot indicate which STARC metrics work best in practice.
2. The theoretical results assume that $\mathcal{S}$ and $\mathcal{A}$ are finite, which are not applicable to continuous environments.

**Questions:**

1.	In Definition 1 and the entire paper, the phrase “differ by” frequently occurs. But what do you mean by it? Say, “$C(R)$ and $R$ differ by potential shaping and $S’$-distribution” in Definition 1, do you mean that we can get $C(R)$ by first potential shaping $R$ and then $S’$-distribution? Does the order matters?
2.	Considering Definition 3, you define $s(R)=c(R)/n(c(R))$ and claim in the following sentence that $s$ would reduce rewards that have the same ordering of policies to the same one. Could you please give a simple example (MDP) and illustrate the calculation process of your defined STARC metric and verifies your claim.
3.	In proposition 2, what is the reward function when calculating $V^\pi$?
4.	In the proof of Proposition 9 and 10, you just prove that “that $c(R_1)=c(R_2)$ if $R_1$ and $R_2$ differ by potential shaping and $S’$-redistribution”. However, in the definition of canonicalization function, a canonicalization must satisfy that “$c(R_1)=c(R_2) $if and only if $R_1$ and $R_2$ only differ by potential shaping and $S’$-redistribution”. Why not consider “only if”?


I am willing to raise my scores if you could solve my concerns.

---

> ### Author Response · Authors · 2023-11-13
> **Response**
>
> We thank reviewer JNkN for their thoughts and feedback! Our answers to your questions are as follows:
>
> Weaknesses:
> 1. It is true that our theoretical results cannot indicate which STARC metric would work best in practice, but our experiments suggest that the $L_1$-norm may be the best choice for both $m$ and $n$.
> 2. We should note that we expect our theoretical results to generalise to (sufficiently well-behaved) continuous environments, as also indicated by our experiments. However, we think it is reasonable to consider this extension of our analysis to be out of scope for this paper.
>
> Questions:
> 1. Yes, that is correct; when we say that $R_1$ and $R_2$ differ by potential shaping and $S'$-redistribution, we mean that there is a potential shaping transformation $t_1$ and an $S'$-redistribution transformation $t_2$ such that $R_2 = t_1 \circ t_2(R_1)$. Equivalently, it means that there is a potential function $\Phi$ such that $E[R_2(s,a,S')] = E[R_1(s,a,S') + \gamma \Phi(S') - \Phi(s)]$. This also means that the order does not matter. We will amend the text, to make this more clear.
> 2. We can certainly do this. Consider a Bandit MDP with several actions and one state $s$, let $c$ be the VAL-canonicalisation for the uniformly random policy, and let $n$ be the $L_2$-norm. In this case, two reward functions induce the same ordering of policies if and only if they differ by an affine transformation. Moreover, for any reward function $R$, we have that $V^\pi(s) = E[R(s,A,s)]/(1-\gamma)$, where $\pi$ is the uniformly random policy, and $A$ is sampled uniformly at random. This means that $c(R)(s,a,s) = R(s,a,s) - E[R(s,A,s)]/(1-\gamma) + \gamma E[R(s,A,s)]/(1-\gamma) = R(s,a,s) - E[R(s,A,s)]$, where $A$ is sampled uniformly at random. In this case, it is easy to see that $c(R_1) = c(R_2)$ if and only if there is a constant $b$ such that $R_2 = R_1 + b$. Division by $n(c(R))$ normalises the resulting reward. This means that $s(R_1) = s(R_2)$ if and only if there is a positive constant $a$ such that $c(R_2) = a \cdot c(R_1)$. Since $c$ is linear, this means that $c(R_2) = c(a \cdot R_1)$. Putting this together, we get that $s(R_1) = s(R_2)$ if and only if $R_2 = a \cdot R_1 + b$ for some $a$ and $b$, which in turn is the case if and only if $R_1$ and $R_2$ induce the same ordering of policies.
> 3. $V^\pi$ should be calculated using the reward function that is being canonicalised. I.e., when calculating $c(R_1)$, $V^\pi$ should be calculated using $R_1$, and when calculating $c(R_2)$, $V^\pi$ should be calculated using $R_2$, etc. We will clarify this in the text.
> 4. It is correct that it should be "if and only if", and we will amend the proofs to make this explicit. Note that if $R$ and $c(R)$ differ by potential shaping and $S'$-redistribution for all $R$, then $R_1$ and $c(R_1)$ differ by potential shaping and $S'$-redistribution, and likewise for $R_2$ and $c(R_2)$. Then if $c(R_1) = c(R_2)$, we can combine these transformations, and obtain that $R_1$ and $R_2$ also differ by potential shaping and $S'$-redistribution.
>
> We hope that these points help to clarify our paper, and that the reviewer will consider increasing their score!

---

> > ### Comment · Reviewer_JNkN · 2023-11-18
> > **Concerns Remain**
> >
> > Thanks to your response. However, I still have concerns remaining to be unsolved.
> >
> > > Yes, that is correct; when we say that $R_1$ and $R_2$ differ by potential shaping...
> >
> > Why does the order not matter? Here I mean the order of potential shaping and $S'$-redistribution. In your response, you said $R_2 = t_1 \circ t_2 (R_1)$, which seems that we should first do potential shaping and then $S'$-redistribution. Please make it clear since the phrase "differ by" is used many times in the paper but not formally defined.
> >
> > > We can certainly do this. Consider a Bandit MDP...
> >
> > Please give proofs in your example if you say something like "only if". Moreover, is this bandit with only one state too special to be a good example?
> >
> > > It is correct that it should be "if and only if", and we will amend the proofs to make this explicit...
> >
> > If possible, please update your manuscript and upload a revised version.

---

> ### Author Response · Authors · 2023-11-18
> **Response**
>
> > Why does the order not matter?
>
> If it is possible to produce $R_2$ from $R_1$ by first applying potential shaping and then applying $S'$-redistribution, then it is also possible to do this by first applying $S'$-redistribution, and then applying potential shaping, and vice versa. To see this, note that we can produce $R_2$ from $R_1$ by potential shaping if and only if $R_2 = R_1 + F$, where $F : S \times A \times S \to \mathbb{R}$ is a reward function such that $F(s,a,s') = \gamma \Phi(s') - \Phi(s)$ for some potential function $\Phi$. Similarly, we can produce $R_2$ from $R_1$ via $S'$-redistribution if and only if $R_2 = R_1 + G$, where $G : S \times A \times S \to \mathbb{R}$ is a reward function such that $\mathbb{E}_{S' \sim \tau(s,a)}[G(s,a,S')] = 0$. That means that we can produce $R_2$ from $R_1$ by a combination of potential shaping and $S'$-redistribution if and only if we can express $R_2$ as $R_1 + F + G$, where $F$ and $G$ satisfy the conditions above. Since $R_1 + F + G = R_1 + G + F$, the order does not matter. We will clarify this in the main text.
>
> > Please give proofs in your example if you say something like "only if".
>
> In each case in that example, the "only if" parts follow from very straightforward though somewhat tedious algebra. Do you think it would be beneficial to provide this in full?
>
> > Moreover, is this bandit with only one state too special to be a good example?
>
> Perhaps, but if we add several states then it will quickly become very complicated to directly determine neccessary and sufficient conditions for two reward functions to induce the same policy order, except by simply appealing to Theorem 2.6 in Skalse & Abate, 2023. This would make the example less illuminating, to the point that it may be easier to simply refer to the proof in Skalse & Abate, 2023, together with Definition 1 and 3 in our paper.
>
> > If possible, please update your manuscript and upload a revised version.
>
> We will upload an amended manuscript which incorporates all the feedback from the reviewers in a few days [EDIT: we have now done this]. As for the proofs of Proposition 9 and 10 specifically, note that all which has to be done is to add the two sentences from point 4 in our previous response.
>
> We hope that this clarifies your questions, and that you will consider increasing your score!

---

> > ### Comment · Reviewer_JNkN · 2023-11-22
> > **Thanks to Your Reponse**
> >
> > Thanks to your response. Most of my concerns have been solved, and I have updated my score.

---

### Official Review · Reviewer_uyUs · 2023-11-01

**Soundness:** 3 good
**Presentation:** 4 excellent
**Contribution:** 3 good
**Rating:** 6
**Confidence:** 4

**Summary:**

The authors motivate the need for metrics quantifying the differences between reward functions without having to compare policies optimized on those rewards. Comparison is provided to previous work, namely EPIC and DARD. The authors define "canonicalization function" as a collapsing of reward functions equivalent under potential shaping and S'-redistribution. Interestingly C_EPIC and C_DARD are not canonicalization functions under this distribution because they do not consider S'-redistribution. STARC is then introduced as a class of metrics defined by a canonicalization function, norm, and metric satisfying various properties. STARC metrics crucially are defined relative to the environment, meaning they are implicitly parameterized by S, A, T, lambda, and the initial state distribution. Value-Adjusted Leveling (VAL) is provided as one example construction of a canonicalization function. The paper proves that all STARC metrics are both sound and complete, providing both upper and lower bounds on worst-case regret, and that any sound and complete pseudometrics are bilipschitz equivalent to STARC metrics. Experiments are conducted on both small MDPs, where strong correlation between STARC-VAL and regret is demonstrated, and in a MuJoCo environment.

**Strengths:**

There is an appreciable amount of novel theory in this work. The definition of canonicalization function simply adds S'-redistribution to the list of invariances compared to EPIC and DARD, but there is novelty in the definition of VAL, definitions of sound and complete pseudometrics in reward space, and theorems stating that STARC is both sound and complete.

The paper provides a satisfying comparison between STARC, EPIC, and DARD, and convincingly explains why EPIC and DARD are not both sound and complete.

The experiment in the small MDP is a convincing demonstration of the superiority of STARC-VAL in practice over other baselines (and also that canonicalization is crucial). The experiment in Reacher demonstrated that STARC-VAL produced the correct (expected) ranking of rewards.

The work is significant because reward function evaluation separate from policy optimization is an important issue in the field, and the work demonstrates that existing methods have issues which STARC resolves.

The paper is well-written and very clear.

**Weaknesses:**

One substantial weakness I see with the reward comparison literature in general is its applicability: the paper is missing a discussion of how reward comparison might be used in practice, when there is no access to the ground truth reward. Is it meant to be used to compare two learned rewards? How does the value influence a ML system designer's decision on what to do next?

In section 2.3, the authors argue for why they think the dependence on T is meaningful. I think it is missing a discussion of the sim2real setting. Does it require a simulator with the true transitions, or simply samples from the true transitions? If the former, I would argue it is a meaningful limitation for many real world tasks due to the sim2real gap.

In Reacher, the authors acknowledge that estimating the metric via sampling involves summing over absolute values which makes all noise positive, thus inflating the metric for PotentialShaped which should have 0 distance. Indeed in table 1, the distance values are not linearly related to what I'd want the metric to say. The authors argue it is not problematic for ranking, which I agree with, but it is crucial for absolute evaluation (as opposed to ranking), which I think is also important. The paper is missing a discussion of absolute evaluation, as well as a discussion of how big an issue they expect this to be for environments in general.

The authors claim to provide a complete answer to the question of how to measure distance between reward functions, but I'd argue completeness requires providing a tight bound. We're not sure how large the spread L to U is.

The authors also acknowledge the theory assumes finite S and A and uses a strong definition of regret.

**Questions:**

- typo on pg 3? definition of pseudometric: should be n(x-v)
- would be great to provide (even 1-sentence) intuition for EPIC (explain the various normalization terms that EPIC paper explains), because it is crucial for understanding your definition of canonicalization function
- how are D_S and D_A chosen in EPIC?
- can you give a brief explanation/intuition for S'-distribution? since it is the difference between your definition of canonicalization function and that used in EPIC
- in def 1: where do R, R_1, R_2 come from? is it all such R's in the reward class?
- can you provide intuition for Im(c)? it appears somewhat out of nowhere.
- in def 1 and 4: should it say differ *only* by (potential shaping and S'-redistribution)?
- why is it not given that minimal CFs exist for any given norm or are unique?
- how should norm and distance metric be chosen in practice?
- how do you choose the policy for VAL canonicalization? why uniform for reacher?

---

> ### Author Response · Authors · 2023-11-13
> **Response**
>
> We thank reviewer uyUs for their feedback! We have responded to your questions and concerns below:
>
> Weaknesses:
> 1. The intended use for these metrics is to *evaluate* reward learning algorithms, in a setting where the true reward function *is* known. For example, we may randomly generate reward functions, use a reward learning algorithm, and then measure how different the learnt reward function is from the true reward function. Alternatively, they can also be used in theoretical work, where we may wish to bound the distance between the true reward and the learnt reward (given different amounts of training data, for example). Such analysis would then indicate how reliable a given reward learning method is, which in turn then tells us how accurate we should expect a learnt reward function to be, in a setting where the true reward function is not known. Note that Gleave et al is cited 37 times, and Wulfe et al is cited 6 times; these references will provide examples of practical situations where reward function pseudo-metrics can be used. For example, see eg Adeniji et al, 2023, Figure 1 in Sanghvi et al, 2021, or Figure 4(c) in Liang et al, 2022.
> 2. We do not require a simulator with the true transitions --- samples from the true transitions are sufficient. For example, the VAL canonicalisation can be estimated using samples. This is discussed in the first paragraph of Section 2.3.
> 3. We agree that this may be a limitation in some contexts. However, note that the error can be reduced by increasing the number of samples used in the estimation. Moreover, it can be eliminated completely if the STARC metric is computed exactly (although this may not be feasible in large environments).
> 4. We agree that the spread between L and U is important. However, note that any metric which is sound and complete must be bilipschitz equivalent to STARC metrics -- this means that whichever metric has the smallest spread between L and U must be quite similar to STARC metric (and it might be an instance of a STARC metric). Note also that the exact value of L and U depends on the transition function; to see this, consider the second figure in Appendix B (the transition function affects the diameter of $\Omega$).
>
> Questions:
> 1.  Yes, this is indeed a typo, thank you for pointing this out.
> 2.  We describe EPIC in Section 1.2. We can make this description more extensive, to make it easier to understand.
> 3.  In our experiments, $D_S$ and $D_A$ were both uniform for EPIC. We will make sure that this is included in Appendix F; thank you for spotting this omission.
> 4.  $S'$-redistribution is simply any transformation that never changes the expected value of the reward function. For example, suppose we have a nondeterministic MDP where an action $a$ in state $s$ leads to $s_1$ or $s_2$ with equal probability, and that $R_1(s,a,s_1) = 1$, $R_1(s,a,s_1) = 0$. Then $S'$-redistribution could be used to produce a reward function $R_2$ where $R_2(s,a,s_1) = R_2(s,a,s_2) = 0.5$.
> 5.  The conditions in Definition 1 are meant to hold for all reward functions -- it can thus be read as "for all $R$, ..., and for all $R_1, R_2$, ... We will amend the text to make this more clear.
> 6.  $\mathrm{Im}(c)$ is the output of the canonicalisation function, when it is applied to the space of all reward functions. From the definition of canonicalisation functions, we get that this is a linear subspace of $\mathcal{R}$ in which no reward functions differ by potential shaping or $S'$-redistribution. A geometric description of this is given in Appendix B, but we can also describe this in more detail in the main text.
> 7.  Yes. We will amend the text to make this clear.
> 8.  For example, relative to the $L_1$-norm or the $L_\infty$-norm, there will be multiple minimal canonicalisation functions. This is because the unit balls of these norms are not strictly convex.
> 9.  We expect that it in practice often will be best to pick $n$ and $m$ so that they can be computed easily. Our experiments suggest that the $L_1$-norm is the best (unweighted) norm for both $n$ and $m$.
> 10. VAL is a valid canonicalisation function for any choice of policy, so we simply used a policy for which $V^\pi$ would be easy to learn. The benefit of using a uniform policy, rather than some deterministic policy, is that exploration then is built in automatically.
>
> We hope that this clarifies our paper, and that the reviewer might consider increasing their score.

---

> > ### Comment · Reviewer_uyUs · 2023-11-23
> >
> > thank you for the responses to my questions and identified weaknesses, they have been addressed.

---

### Official Review · Reviewer_9CDF · 2023-11-01

**Soundness:** 3 good
**Presentation:** 3 good
**Contribution:** 3 good
**Rating:** 6
**Confidence:** 3

**Summary:**

The paper introduces STARC, a class of pseudo-metric over reward functions. Intuitively, it measures how two reward functions differ by computing a distance between the normalized representative elements of their equivalence classes with respect to potential shaping and S'-redistribution. Theoretical results are proven for STARC, which is also contrasted with previous propositions.

**Strengths:**

The proposed class of pseudo-metrics is novel and natural. It enjoys nice theoretical properties such as soundness and completeness, which are not satisfied by previously-proposed pseudo-metrics.

Concrete examples of instances of such pseudo-metrics are provided.

The paper is well-written and clear.

**Weaknesses:**

The premise of this work is that the proposed pseudo-metric could be used to quantify how a learned reward function differs from the true one. Unfortunately, in practice, when one needs to learn a reward function, the true one is not known, which makes the usefulness of such pseudo-metric not clear to me.

The interchangeable use in the text of pseudo-metric and metric makes things a bit confusing sometimes. I think it would be better to use consistently pseudo-metric to refer to pseudo-metric.

Minor:

The definition of metric m from norm n contains a typo (page 3).
The definition of the L_p norm is missing the absolute value.

Definition 6: there exists two -> there exist two

The following paper:
Joar Skalse, Nikolaus H. R. Howe, Dmitrii Krasheninnikov, and David Krueger. Defining and characterizing reward hacking
appears twice in the references.

**Questions:**

The authors first write:
"STARC metrics in practice CAN have a much tighter correlation with worst-case regret than both EPIC and DARD"
then continue with:
"This means that STARC metrics both attain better empirical performance"
Is it this always true?

What does this sentence mean? "the property that increasing R1 cannot decrease R2"

Could you provide some examples where the true reward is not known, but such pseudo-metrics could be helpful?

---

> ### Author Response · Authors · 2023-11-13
> **Response**
>
> We would like to thank reviewer 9CDF for their review and feedback!
>
> We should clarify that the intended use for these metrics is to *evaluate* reward learning algorithms, in a setting where the true reward function *is* known. For example, we may randomly generate reward functions, use a reward learning algorithm, and then measure how different the learnt reward function is from the true reward function. Alternatively, they can also be used in theoretical work, where we may wish to bound the distance between the true reward and the learnt reward (given different amounts of training data, for example). Such analysis would then indicate how reliable a given reward learning method is, which then tells us how accurate we should expect a learnt reward function to be, in a setting where the true reward function is not known. Note that Gleave et al is cited 37 times, and Wulfe et al is cited 6 times; these references will provide examples of practical situations where reward function pseudo-metrics can be used. For example, see eg Adeniji et al, 2023, Figure 1 in Sanghvi et al, 2021, or Figure 4(c) in Liang et al, 2022. We will amend the introduction, to make this intended use case more clear.
>
> We will amend the text to consistently use the term pseudo-metric, to make it less ambiguous. We also fix the typos that you have spotted -- thank you for pointing them out to us!
>
> Questions:
> 1. Our experiments suggest that this is not always true, but that it is true for most reasonable choices of normalisation norm and metric --- see Figure 1.
> 2. This means that if two reward functions $R_1, R_2$ and two policies $\pi_1, \pi_2$ satisfy that $J_1(\pi_1) > J_1(\pi_2)$, then it must be the case that $J_1(\pi_1) \geq J_1(\pi_2)$. In other words, if the expected cumulative sum of $R_1$-reward increases when we go from $\pi_2$ to $\pi_1$, then it cannot be the case that the expected cumulative sum of $R_2$-reward decreases. We will amend the text, to make this more clear.
> 3. We can evaluate a reward learning algorithm in a setting where the true reward is known. The result of this evaluation would then inform how much we should trust that algorithm, in a setting where the true reward is not known.
>
> We hope that this will help to clarify our paper, and that the reviewer might consider increasing their score.

---

> > ### Comment · Reviewer_9CDF · 2023-12-04
> >
> > Thank you for your responses. This addresses my concern about the usefulness of the proposed approach.

---

### Author Response · Authors · 2023-11-19
**Revised Manuscript**

We have now uploaded a revised version of our manuscript, which incorporates the current feedback from the reviewers.

The changes we have made include:
1. We have replaced each occurence of "metric" with "pseudometric", except when we mean to specifically refer to a (proper) metric, or when "metric" is used as part of the name of a pseudometric (so, we still write "the EPIC metric", etc).
2. We have fixed the typo in the definition of the metric that is induced by a norm.
3. We have fixed the typo "there exists two".
4. We have fixed the references to the paper "Defining and Characterising Reward Hacking", by Skalse et al.
5. We have replaced "increasing $R_1$ cannot decrease $R_2$" with less ambiguous terminology.
6. We have included a few sentences that provide more intuition for the definition of EPIC.
7. We have clarified how Definition 1 quantifies over reward functions.
8. We have described which $\mathcal{D}_\mathcal{S}$ and $\mathcal{D}_\mathcal{A}$ were used for EPIC and DARD in the experiments.
9. We have commented on why we chose to use the uniform policy for VAL in our experiments.
10. We have clarified that Definition 1 and 4 mean that the reward functions differ only by potential shaping and $S'$-redistribution.
11. We have provided some comments with intuition for $\mathrm{Im}(c)$.
12. We have clarified which reward is used in the computation of $V^\pi$ in the definition of the VAL canonicalisation.
13. We have amended the proofs of Proposition 9 and 10, to explicitly spell out the "only if"-direction.
14. We have clarified what it means for two reward functions to "differ by potential shaping and $S'$-redistribution".
15. We have included an appendix with tables that describe the individual impact of the choice of $m$ and $n$ when the canonicalisation function is held constant.
16. We have included more information in Section 4.1, so that a list of all canonicalisation functions used in the experiment now is included in the main text.

To make space for these edits, we have moved Section 3.2 (Issues With Epic) to the Appendix.

---

### Meta-Review · Area_Chair_SfcB · 2023-12-06

**Metareview:**

The paper proposes STARC, a new class of pseudo-metrics on the space of reward functions, thereby allowing us to quantify differences between reward functions. STARC pseudo-metrics satisfy several valuable properties and could be potentially useful for both the theoretical and empirical analysis of reward learning algorithms. The reviewers agreed that the paper addresses an important problem of quantifying differences in reward functions and acknowledged the novelty of the work. However, the reviewers also raised several concerns and questions in their initial reviews. We want to thank the authors for their responses and active engagement during the discussion phase. The reviewers appreciated the responses, which helped in answering their key questions. The reviewers have an overall positive assessment of the paper, and there is a consensus for acceptance. The reviewers have provided detailed feedback, and we strongly encourage the authors to incorporate this feedback when preparing the final version of the paper.

**Justification For Why Not Higher Score:**

The paper could possibly be given a score of "Accept (spotlight)" based on score calibration with other accepted papers.

**Justification For Why Not Lower Score:**

"Accept (poster)" is justified as the reviewers have an overall positive assessment, and there is a consensus for acceptance.

---

### Decision · Program_Chairs · 2024-01-16

Accept (poster)